# Revealing Key Dimensions Underlying the Recognition of Dynamic Human Actions
André Bockes [1], Martin N. Hebart[2,3,4] & Angelika Lingnau [1,4] ✉

How do we understand actions performed by others? Recent studies suggest that actions can be represented as points in a multidimensional space, where the perceived similarity of two actions is thought to be related to their proximity within this space. Here we present a data-driven approach to reveal key dimensions underlying this space using a carefully selected stimulus database of 768 one-second video clips spanning 256 action categories. We gathered similarity ratings for these videos from 6,036 participants and used a computational modeling procedure to identify key dimensions underlying these ratings. This approach revealed 28 meaningful dimensions (e.g. interaction, sport and craft) which capture information concerning human actions as well as a broad range of related domains (e.g. living and non-living things). Explicit ratings of actions along these dimensions gathered in a separate group of participants revealed a high correlation between ratings and weights along these dimensions, demonstrating that these dimensions are interpretable and can be used by participants. The multidimensional action space established in the current study enables the quantification of the similarity between different actions, which will be useful for the generation of hypotheses and future experimental manipulations. Together, our results provide a window into the nature of the representations underlying the ability to interpret other people's actions and pave the way for future lines of research.

In daily life, we are constantly confronted with the need to interpret other people's actions, for example, when exercising together, during a conversation, or on our way to work. A key unresolved issue is to figure out according to which principles we can classify actions despite the multiple different ways in which they can be performed (for reviews, see e.g., refs. [1–5]). Building on previous work on mental representations of objects[6–8], several authors have proposed that actions, like objects, can be described in a multidimensional space, where each point in the space corresponds to one action[3,9–15]. The closer two actions are within this representational space, the more similar they are thought to be in terms of their subjective similarity. Following this idea, a key step towards addressing the question of how we understand other people's actions is thus to determine the dimensions underlying this multidimensional space.

Given the wide range of actions we encounter in our daily lives, it is far from clear which dimensions are most relevant to recognize and distinguish between actions. Several recent studies addressing this question have reported a small set of dimensions, such as tool-relatedness, food-relatedness and valence[9,10,12–14,16,17]. While some dimensions appeared to be consistent across studies (e.g., food-relatedness), other dimensions were less

stable (e.g., manner of motion), likely because a limited range of actions was included (but see ref. [12], for verbal material), because types of stimulus materials varied (verbal material, static images, videos), and because the analysis approach to examine dimensions varied from hypothesis-driven to various data-driven approaches. Moreover, previous attempts were insufficient to explain behavior and, therefore, provided only an incomplete characterization of properties that are relevant for the ability to recognize and categorize actions. Consequently, we still have a surprisingly limited understanding regarding (a) the key dimensions underlying the multidimensional space of actions and (b) the degree to which these dimensions are related to behaviorally derived action similarity judgements obtained in a separate group of participants.

To address this gap, we combined a data-driven approach to reveal key dimensions underlying the representational space of actions with the construction and evaluation of a large data set of actions depicted as short videos. Using this approach, we aimed to (1) determine dimensions underlying the representational space of a wide range of actions and to (2) examine whether these dimensions are interpretable and meaningful to a degree that they can be used by human participants.

[1]Chair of Cognitive Neuroscience, University of Regensburg, Regensburg, Germany. [2]Computational Cognitive Neuroscience and Quantitative Psychiatry, Justus Liebig University, Gießen, Germany. [3]Center for Mind, Brain and Behavior, Universities of Marburg, Gießen, and Darmstadt, Marburg, Germany. [4]Vision and Computational Cognition Group, Max Planck Institute for Human Cognitive and Brain Sciences, Leipzig, Germany. ✉e-mail: angelika.lingnau@ur.de

To address these questions, we created and validated a video database containing 768 naturalistic 1-s video clips from 256 human action categories and gathered ratings from 6036 participants through a crowd-sourced triplet odd-one-out experiment.

## Methods
This study has not been preregistered.

### Participants
All data were collected following all relevant ethical regulations and rules of the ethics committee of the University of Regensburg (Protocol 21-2603-101). Gender information was provided by participants. No data was collected on race or ethnicity.

### Amazon Mechanical Turk workers
A total of 6036 workers (2807 female, 3221 male, 8 other, mean age: 35.8 years) from the online crowdsourcing platform Amazon Mechanical Turk participated in the triplet odd-one-out experiment. All workers were based in the United States and had already successfully completed over 100 other tasks on the platform Amazon Mechanical Turk (acceptance rate: >95%). All workers provided informed consent.

After data collection, following Hebart et al.[7], we conducted additional data filtering (see Section 'Data analysis' for details), resulting in a final dataset comprising 3905 individual workers (1942 female, 1956 male, and 7 other, with a mean age of 36.9 years). Workers were compensated financially for their time.

### Laboratory participants
A total of 135 participants (114 female, 20 male, 1 other, mean age: 23.4 years) recruited from the University of Regensburg were invited to take part in three additional experiments: (1) a total of 94 participants (88 female, 5 male, 1 other, mean age: 21.6 years) took part in the Stimulus validation; (2) 20 participants (12 female, 8 male, mean age: 29.5 years) took part in a Pilot study preceding online data collection, and (3) 21 participants (14 female, 7 male, mean age: 25.5 years) took part in the Dimension labeling and the Dimension rating experiment following the main experiment. All laboratory participants provided written informed consent. The sample size for (3) was based on Hebart et al.[7].

### Human Action Video Dataset
The Human Action Video Dataset used for all experiments was extracted from the Moments in Time (MiT) dataset[18]. This was achieved through a multi-step selection and editing process (see Fig. 1) described in more detail in the following section. In brief, from over one million individual 3-s videos, spread over 339 different categories, we systematically selected a final set of 768 individual videos representing 256 human action categories, with three examples per category, carefully inspected by all three authors, and cut to 1-s length.

### Multi-step selection and editing process
First, a subset of approximately 2500 videos was hand-selected from the training and validation set of the MiT dataset (see next section for additional details). Second, we aimed at cutting the original videos to one second segments, allowing us to gather behavioral data in a more efficient way and avoiding scene cuts and transitions (see also Fig. 1b). To this end, we sought to identify the most informative segment within each video using a deep neural network (ResNet-50[19]), which was pretrained on ImageNet[20] and fine-tuned on the MiT video data[18]. The model was assessed via the MiT webpage (http://moments.csail.mit.edu/). After standardizing the frame rate for all videos of the subset, we extracted, separately for each frame in each video, the prediction accuracy and the corresponding softmax vector—a vector containing the classification probabilities across all 339 categories. This information was used to identify the individual frame per video displaying the highest prediction accuracy. Third, this frame was used to help guide and cut the optimal 1-s segments from the original videos. Note that

during this procedure, sound was removed from the original MiT videos. Therefore, no auditory information was presented during subsequent experiments. For all videos, an equal resolution and 4:3 aspect ratio were established.

### Initial selection of action videos from the MiT dataset
The MiT dataset contains many different action categories. After excluding categories depicting non-human actions or events (e.g., 'dripping' depicting only moments of water dripping), visual inspection revealed a potential relative overrepresentation of certain categories (e.g., sport). To yield a more balanced distribution of categories, we used a clustering approach based on the softmax vectors extracted from the ResNet-50. Specifically, to avoid over- or underrepresenting of action categories (for example, potentially including 50 action categories loosely representing sports but only three categories representing interactive actions), we analyzed the distribution of action categories within the dataset using the softmax vectors extracted from the ResNet-50. Each action category contained several videos. Each video in that category contained 90 frames. A softmax vector was obtained for each of those frames. The softmax vector showing the highest classification accuracy for the video's respective correct category was selected. Next, these vectors were averaged over all videos within a given category. This created a representative vector for each action category. Then, we constructed a dissimilarity matrix by computing pairwise cosine distances between these representative vectors. This matrix was submitted to a hierarchical clustering analysis. We used the resulting dendrogram structure and its corresponding cluster lists to identify meta-clusters of action categories. To balance the internal representation of all categories, we aimed for meta-clusters that contain approximately the same number of categories. One meta-cluster showed a number of diverse categories notably higher than the other meta-clusters (>10 instead of on average 2.3 categories per cluster). Therefore, we carefully removed some categories from this cluster. Further, we removed very specific sport-related terms (like 'putting') and merged age- and gender- specific categories (e.g., 'adult + female + singing', 'adult + male + singing' and 'child + singing'). After additional stimulus validation (see next section), a total of 256 action categories was selected as the final category set. For each of those 256 action classes, three representative exemplars were hand-selected, resulting in a total of 768 stimuli. They were evenly distributed across action categories to reduce any possible representative redundancy.

### Stimulus validation
To ensure that participants were able to recognize the actions depicted in the one-second videos, the stimulus material was evaluated using a multi-step rating study on the online platform SoSciSurvey (https://www.soscisurvey.de) in a total of $N = 94$ participants. $N = 65$ participants were presented consecutively with static representations of the 1-s video clips and were instructed to rate their typicality on a seven-item Likert scale. To that aim, the stimulus was depicted at the top of the browser window. The depicted action was named in the instructions, and participants were asked to rate how well the action was depicted in the video on a scale from one ('very untypical') to seven ('very typical'). Moreover, to determine whether participants were able to recognize the action, for each stimulus, participants were presented with up to 18 different labels and had to identify the correct label. The resulting naming accuracies were used to remove action categories from the dataset (exclusion of categories if the average naming accuracy over all three exemplars per category and over all participants was <0.7, and at least two of the three exemplars per category showed a naming accuracy <0.7). Note that due to a filing error, four categories escaped the exclusion as they were filed as displaying only one exemplar with an accuracy rating <0.7, while in reality showing a naming accuracy <0.7 in two or three exemplars.

Our multistep selection process described above led to the inclusion of 256 human action categories in the final Human Action Video Dataset (see also Supplementary Table 1). Additionally, a group of $N = 29$ participants

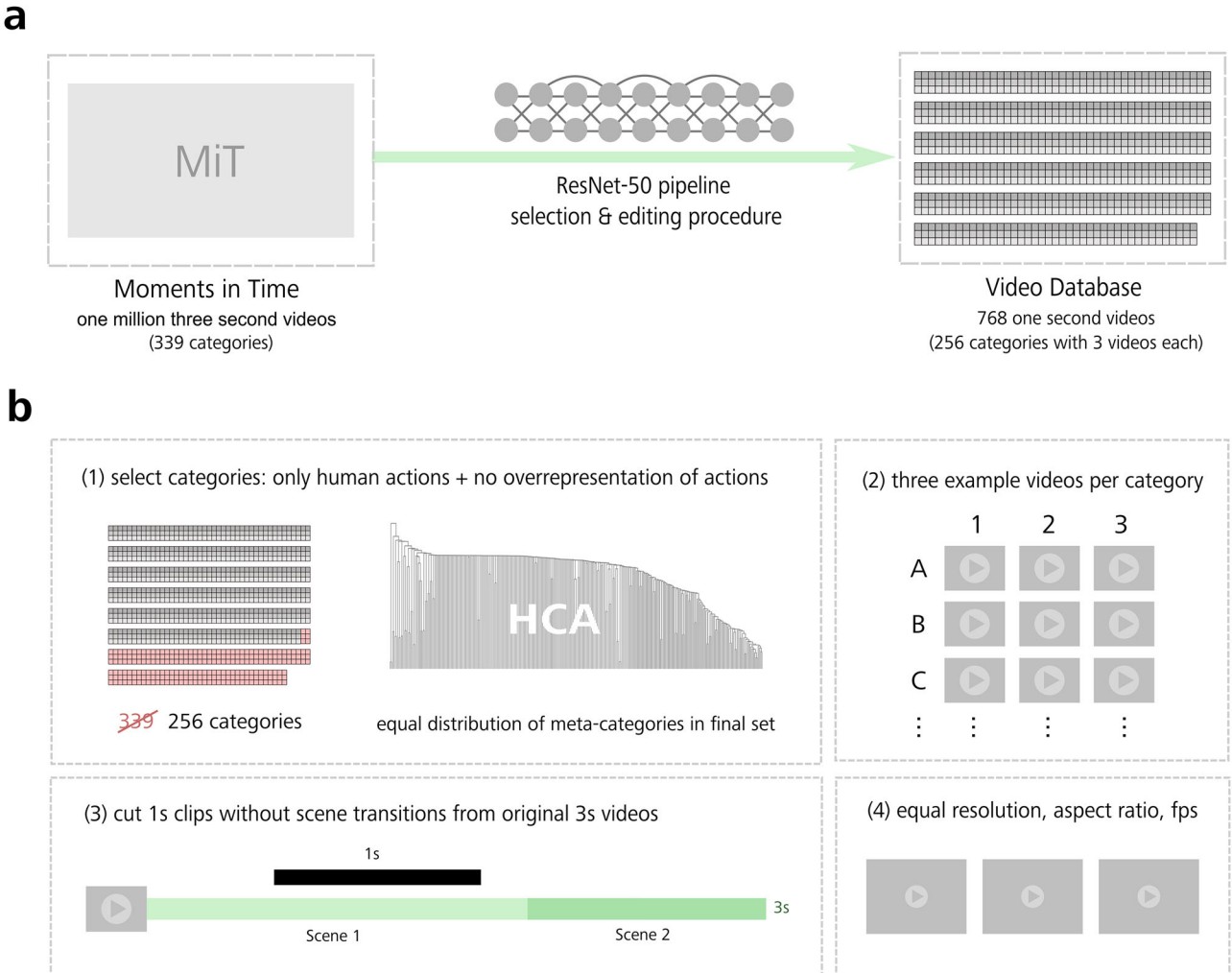

**Fig. 1 | Creating the Human Action Video Database. a** Starting with the Moments in Time database[18], we used a multi-step process to generate a naturalistic, well-balanced dataset of 1 s videos, consisting of 768 videos from 256 action categories. To identify representative 1 s video segments and assess the similarity structure of the action category distribution within the dataset, we used a deep neural network (DNN), combined with a human-guided selection process. **b** To generate the database, we (1) removed non-human action categories from the initial database and then used the softmax vectors from a DNN fed with hand-selected human-related action videos for a hierarchical clustering analysis (HCA, see corresponding dendrogram), identifying 256 categories, (2) manually selected three individual video stimuli to represent each action category, evaluated with a human rating experiment for understandability and interpretability of the videos, (3) selected a representative 1 s video segment additionally eliminating scene cuts, and (4) standardized resolution, aspect ratio and frame rate (fps) for all videos.

was used to evaluate the general recognition differences between the static and dynamic stimulus versions based on a subset of the stimulus material.

## Data collection

**Pilot study: data collection**. Before the main data collection was executed online, we carried out a laboratory pilot experiment to (1) test the feasibility of the triplet odd-one-out task, (2) evaluate data quality, and (3) identify strategies used by the participants. $N = 20$ participants underwent the triplet odd-one-out task procedure, using the same Amazon Mechanical Turk Web Interface as used during the main data collection. The study was followed by a short questionnaire and a brief interview. Participants, recruited from the University of Regensburg, completed 20 sets of 32 trials each. Triplets for these trials were sampled using a subset of 48 randomly chosen videos from the main stimulus set. During sampling, 50% of trial sets were unique to each participant. The other 50% of trial sets were shared among all participants to control for interpersonal differences and choice consistency among participants. For each participant, trial sets alternated between individual and shared. In total, 6720 unique triplets were sampled for the 48-video subset. This resembles 38.9% of all possible combinations using the 48 subset-videos

(48 choose 3 = 17,296 possible combinations). Overall choice consistency in the 20 laboratory participants was 69.59%.

**Triplet odd-one-out experiment: triplet sampling**. The triplet odd-one-out task consisted of selecting one out of three presented videos. Therefore, data sampling included choosing non-repetitive triplets of videos that were presented to participants. A total of 75.2 million possible combinations of exclusive triplets can be created out of the 768 video stimuli (768 choose 3, ≈75.2 million possible combinations, not including positional permutations). The number of trials was determined on the basis of both available resources and feasibility, yet aiming for the highest possible data quality based on estimated choice consistency (see also ref. 7). In total, we collected 1,252,704 odd-one-out choices, including 1,186,795 unique video triplets. These unique triplets reflect 1.58% of the possible combination matrix using all 768 individual video stimuli. The number of triplets was found to be sufficient with respect to overall choice consistency (see section 'Data analysis' for details).

The 1,252,704 odd-one-out combinations were split into 39,147 sets of 32 trials each. This data included the creation of (1) two repetitions of all 17,296 possible triplet combinations for a subset of 48 randomly chosen

video stimuli (to compare main cohort choice behavior against choice behavior from pilot study participants (see section 'Participants/Laboratory participants')) to evaluate inconsistencies (see section 'Data analysis/Pilot study: data collection')), (2) 40 repetitions of 1000 randomly chosen triplets (to estimate general choice consistency for quality control), and (3) the main data of 1,178,112 triplets. The main data was sampled by creating a two-dimensional matrix of all 768 video stimuli and drawing from each matrix cell four times. To create four distinct triplets for each cell, each time a different third stimulus was chosen. Due to repetitions of individual triplets during the whole sampling procedure, the final number of genuinely unique sampled triplets was 1,186,795. The positioning of the videos (left, center, right) was chosen at random without further control of positional permutation. For the 48 video stimuli where the matrix was sampled exhaustively, individual triplets were distributed across the entire dataset at random.

**Triplet odd-one-out experiment: task and data collection.** The triplet odd-one-out task was carried out in sets of 32 trials. Workers were able to choose how many sets they would like to take part in. On each trial, workers were shown three videos arranged horizontally next to each other in a browser window. They were asked to use their mouse to click on the video, which was the least similar to the other two videos (see Supplementary Fig. 1 for the instructions shown to participants). The next trial started after an intertrial interval of 200 ms. Data were collected over a period of 20 workdays (spread across 6 consecutive weeks), during a daily time span of four hours between 8:00 a.m. and 12:00 p.m. (PT).

**Dimension naming experiment: task and data collection.** In the dimension naming task, set up in MATLAB (MathWorks, Natick, MA, USA), participants were presented with an individual naming screen, separately for each dimension. Naming screens contained up to 21 different videos arranged on a continuous scale, based on the corresponding model weights of the displayed dimension (see also Fig. 2a). The videos were depicted as static images and could be played individually via mouse click. The scale was divided into seven levels and ranged from very typical (left) to very untypical (right) and not typical at all (far right), each represented by up to three videos (7 scale levels × 3 representative videos per level = 21 videos).

Participants were instructed to explore the range of the scale, represented by the different videos, and to provide up to three different labels for the corresponding dimension. Participants were instructed not only to focus on the high end of the scale but also to take the whole discriminative width into account for labeling the dimension. The order of dimensions to be named was randomized for each participant. Moreover, to minimize possible naming bias, the order in which the up to three videos representing each level of the scale were arranged from top to bottom was also randomized for each participant. Before data collection, to get familiar with the task, participants were provided with a test trial displaying a dummy dimension that was unrelated to the dimensions revealed by the SPoSE modeling procedure ('monetary value of objects involved in action').

**Dimension rating experiment: task and data collection.** The dimension rating experiment (see Fig. 2a) was executed directly after the dimension naming experiment. Participants were instructed to rate a reference video on the same scales that were used for the dimension naming task. Participants were instructed to position the video along the rating scale via mouse click, using the whole width of the scale. Altogether, there were 20 action videos, randomly selected from the video database, which were rated over all 28 dimensions, resulting in 20 × 28 = 560 trials. Videos included the following action categories: ascending, balancing, bicycling, bowing, celebrating, climbing, dusting, exiting, flipping, hitchhiking, licking, picking, playing sports, preaching, sailing, saluting, sitting, slicing, smiling, and tapping. The order of reference videos, as well as the order of dimensions, was randomized across participants. Moreover, the order in which the up to three videos, representing each level of the scale, were arranged from top to bottom was

randomized for each participant. Prior to data collection, participants were provided with a test trial to get familiar with the task.

## Data analysis

**Triplet odd-one-out experiment: data filtering.** Workers were excluded if they exhibited noticeably fast responses in at least three sets of 32 trials (speed cut-off 12.5% or more responses <900 ms and 50% or more responses <1200 ms) or if they exhibited highly deterministic responses (50% or larger chance of choosing one option) in at least six sets of trials. Following this step, all individual trials displaying reaction times below 900 ms were excluded. Finally, 779,467 trials out of the initial total 1,252,704 trials were used for further data analysis, corresponding to 62.2% of the raw data. Note that the percentage of raw data we kept after data filtering is lower than the percentage reported in previous studies (e.g., 72.9%[21]), which might be due to (a) the fact that the odd-one-out task leads to less consistent answers for dynamic action videos than for static images and (b) the overall decrease of data quality observed with data collected via MTurk over the last few years (e.g., ref. 22). Given the overall high choice consistency in the laboratory sample for the 48 video subset (69.59%), this result may speak to the reduced overall data quality on MTurk. With this filtering procedure, we enforced a higher data quality. Based on the 40 repetitions of 1000 randomly chosen triplets (see section 'Data collection'), we determined a choice consistency among participants of 57.32%, where 33.3% represents random choice.

**Triplet odd-one-out experiment: computational modeling procedure.** To generate dimension embeddings, we used the Sparse Positive Similarity Embedding (SPoSE) approach (for details, see refs. 7,23). This method was chosen as it has been successfully applied in the past, which also facilitated efficient comparisons of our results to object dimensions[7]. In brief, this procedure consisted of the following steps. Embeddings were initialized with a 768 × 100-matrix, where 768 represents the number of different video stimuli used for the triplet odd-one-out task and 100 represents the number of initial latent dimensions for the modeling procedure. Thus, each row of the matrix represented the weights over all initial dimensions for one video stimulus, and each column represented the weights for one dimension over all video stimuli (see also Fig. 3). The values within the cells of the matrix illustrate the representation of each video through the individual dimensions and vice versa. This matrix was randomly initialized with values in the range between zero and one. Before the training started, the data for the modeling procedure was separated into a training and a test set using a 90%/10% split.

The optimization procedure followed a stochastic gradient descent implemented in the Adam algorithm[24]. The objective function during this procedure was aimed to minimize the cross-entropy (logarithm of the softmax function) as well as a regularization term based on the L1-normalization. This normalization was enforcing sparsity within the embedding representations. Moreover, a regularization parameter lambda ($\lambda = 0.008$), selected via hyperparameter optimization (see next section for further details), controlled the tradeoff between model complexity and sparsity, controlling the final number of dimension embeddings. Another prerequisite of the model procedure was the enforcing of positivity and, therefore, interpretability of dimension values. A batch size of 128, window-size of 50 and learning rate of 0.001 were used, as suggested in the SPoSE example call (https://github.com/ViCCo-Group/SPoSE). The optimization procedure stopped after a maximum of 500 epochs.

For details on the optimization process, see the 'Results' section 'Odd-one-out task data and modeling procedure' and Fig. 3. In brief, by modeling human selection behavior using the participant's triplet odd-one-out choices, the SPoSE model reduced a randomly initialized dimension space to a limited set of dimensions capturing human action categorization behavior.

The modeling procedure removed those dimensions which did not represent necessary information for the prediction task. To this aim, after completing optimization, all embeddings showing a summed value smaller

**a** rate the action according to its typicality

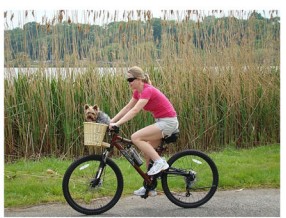

very typical very untypical not at all

**b**

20 randomly sampled action-videos

similarity (human ratings) similarity (model weights)

-0.35 ▬ 1

**c**

$r = 0.78$

similarity (model weights)

similarity (human ratings)

**Fig. 2 | Dimension rating task and results. a** $N = 21$ participants were asked to position 20 different reference action videos (e.g., 'bicycling') on each dimension. The dimension scale was separated into seven levels, ranging from very typical (left side) to very atypical (right side), based on the corresponding model weights of each dimension. Each level was represented by up to three representative videos. The example dimension scale depicts the dimension 'food'. **b** The 20 reference videos were randomly sampled from all available 768 action videos. Left panel: similarity matrix based on the human dimension ratings from $N = 21$ participants, right panel: similarity matrix based on SPoSE model weights corresponding to each of the stimuli. **c** Pearson correlation between the two similarity matrices shown in panel (**b**) (Pearson's $r = 0.78$; $P < 0.001$; randomization test; 95% CI, 0.71–0.83). For this figure, the original MiT video frames were replaced by images under a free-to-use license (Pexels: Askar Abayev, Center for Ageing Better, cottonbro studio, KampusProduction, Miriam Alonso, Roman Odintsov, vitalina, Yan Krukau; Pixabay: brenkee, Michelle_Raponi, PublicDomainPictures; Unsplash: DDP, Dipin Bhattarai, EqualStock, Frames for your Heart, Jovan Vasiljević, Julia Tsukurova, Mary West, Michael Kahn, National Cancer Institute, Rithwick. Pr, Sincerely Media).

than 0.1 were removed, leaving a subset of 28 of the 100 initialized embedding dimensions (see 'Results' section). Dimensions were sorted in descending order with respect to the sum of their weights across all video stimuli.

**Triplet odd-one-out experiment: hyperparameter optimization.** To assess the optimal value for the hyperparameter $\lambda$, which controls model sparsity during the optimization procedure, an optimization search was performed. Thirteen different values were chosen in the range between 0.006 and 0.012 with a step size of 0.0005. Each value was then used for an individual SPoSE optimization process. For each chosen lambda value, ten different random seeds were selected for model initialization. A total of 130 individual model optimizations were conducted. For each lambda value, the average over all ten corresponding seeds was calculated for their optimized SPoSE model's respective validation losses. The one lambda value that, on average, displayed the smallest validation loss was then chosen as a hyperparameter during the main computational modeling procedure.

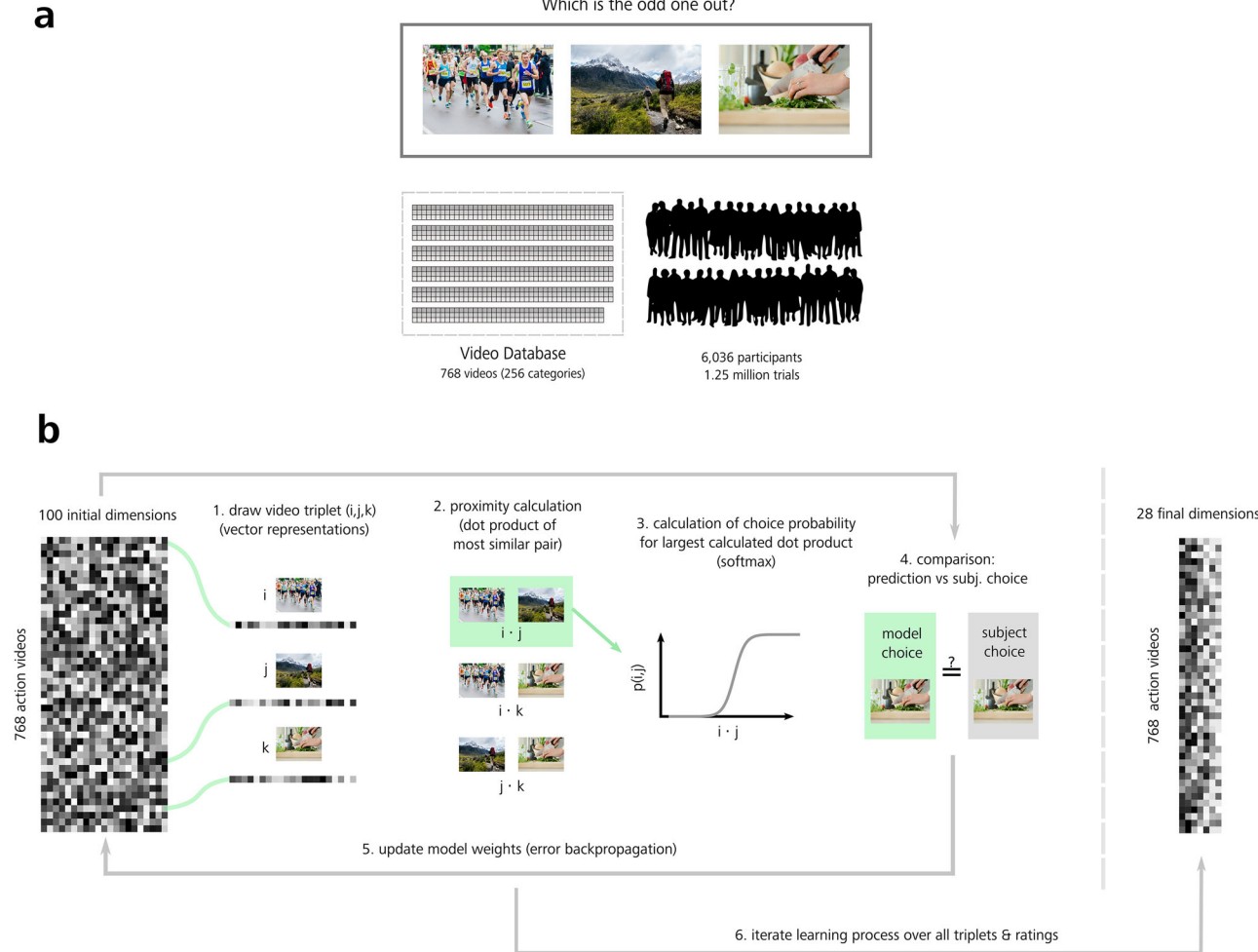

**Fig. 3 | Odd-one-out behavioral task and modeling procedure of dimensions underlying action similarity judgments. a** We collected 1,252,704 odd-one-out choices on stimuli from our Human Action Video Database from 6036 online participants. The number of unique triplets within these choices reflects 1.58% of all possible triplets using all 768 video stimuli. **b** Dimensions were generated using the SPoSE-model[7,23]. First, a 768 (video stimuli, rows) × 100 (starting dimensions, columns) weight matrix was initialized randomly. In an iterative loop, a triplet of three vector representations was drawn from these starting embeddings. Next, for all three pairs of videos, their proximity was calculated as the dot product between their vectors. Then the softmax choice probability for the largest dot product was computed and compared with the odd-one-out choice obtained from participants. Based on this comparison, the model weights were updated using error backpropagation. The model required the final embeddings to be both positive and sparse. Embedding weights containing little to no information were set to zero after training. Hence, the 100 initial dimensions were reduced to 28 final dimension embeddings. For this figure, the original MiT video frames were replaced by images under a free-to-use license (Pixabay: tookapic, Placidplace; Unsplash: Alyson McPhee, Toomas Tartes).

Due to the general stochastic nature of the modeling procedure using the SPoSE approach, for each different random initialization of the weight matrix, potentially slightly different results can be expected in the embedding. To check for reproducibility and the number of resulting embedding dimensions, we ran 20 individual SPoSE modeling processes for the optimal hyperparameter lambda. On average, this approach revealed 26.65 dimensions over the different initializations (see Supplementary Fig. 3).

**Dimension naming experiment: data analysis**. After data collection, smaller typographic and capitalization errors were corrected by the experimenters. As all participants were native German speakers, labels were provided in German. They were translated into English using DeepL (www.deepl.com). Next, to visualize the labels that were provided most frequently, word clouds were constructed using the MATLAB function '*wordcloud*', with text size proportionally reflecting naming frequency. Additionally, the translated label lists were fed into a large language model (ChatGPT-4) using the following prompt: "*Please provide a label that best summarizes the content of the following words. The label should also consider the frequency of the chosen terms.*" The resulting dimension labels are provided in Supplementary Table 2.

**Dimension rating experiment: data analysis**. After data collection, all ratings from the scale level 'not at all' were set to zero; the remaining ratings were scaled to values between zero and one, and data were averaged across participants. Next, to construct a human rating dissimilarity matrix, pairwise Pearson correlations were calculated for each pair of video ratings. In addition, model weights for the 20 corresponding reference videos were extracted from the SPoSE model, scaled to values between zero and one for better comparability and transformed into a similarity matrix also using pairwise Pearson correlation. Finally, the similarity between the human rating similarity matrix and the SPoSE model similarity matrix was determined using the Pearson correlation coefficient. The assumption of metric data was met. Data distribution was assumed to be normal, but this was not formally tested.

Additionally, non-parametric randomization tests on correlations between the human rating representational similarity matrices and the SPoSE model representational similarity matrices were executed using the following three steps: (1) Creating 100,000 similarity matrices via randomly shuffling the reference action video labels, (2) rerunning the correlation with the measured similarity matrix and (3) calculating the p-values as the percentage of permutations at least reaching the true similarity (see also ref. 7).

## Results

The goal of this project was to reveal the dimensions underlying the representational space of observed actions. To achieve this, we had to overcome several obstacles. First, we had to acquire a suitable set of human action stimuli covering a broad range of categories. Second, we had to define an experimental task that is capable of creating enough behavioral rating data to successfully train a computational model extracting dimensions that underlie the ability to judge the similarity of observed actions. Third, we had to acquire data from enough participants for model training and subsequent dimension generation.

To address the first challenge, we reasoned that short video clips of humans are better suited than static images to reveal the set of dimensions underlying our ability to categorize actions. In contrast to objects, actions dynamically unfold in time. Since none of the existing video databases, typically developed in the domain of computer vision, fulfilled our needs, we decided to create a well-controlled video database of naturalistic and diverse video stimuli depicting human actions (see next section and Methods, 'Human Action Video Dataset' for details).

To address the second challenge of defining an experimental task, we decided to use a three-alternative forced choice (3-AFC) task, also known as the triplet odd-one-out task[7,21]. The task consists of three stimuli depicted horizontally next to each other (e.g., a person walking a dog, a group of people running, and a couple preparing a meal together). Participants are instructed to choose the stimulus that appears to be the least suited to the group of three (e.g., the couple preparing a meal in the example provided above). This task is very intuitive to understand and, therefore, allowed us to gather a vast amount of data in a time-efficient manner by continuously presenting new triplets to the participants.

To address the third challenge of gathering data from a sufficiently large group of participants for the triplet odd-one-out task, we chose a crowd-based approach in combination with an online study using the platform Amazon Mechanical Turk (MTurk) (https://www.mturk.com/).

### Creating a Human Action Video Database suitable for crowdsourcing

To be suitable for our approach, the action video database had to fulfill two criteria. First, to be able to extract meaningful dimensions representative of the way we distinguish between actions we encounter in our daily lives, we needed it to cover a wide range of actions. Second, to avoid that dimensions are biased by an overrepresentation of specific actions, we aimed for an equal distribution of actions across a range of different action categories (see Fig. 1).

As a starting point, we used the Moments in Time (MiT) database[18]. While MiT provides a valuable basis, with a large number of 3 s video clips from 339 different categories of actions and events, videos in this database (a) include both human actions and events caused by non-human agents, (b) often contain one or several scene cuts, and (c) are not controlled for frame rate, resolution and aspect ratio.

To select suitable videos from the MiT database while overcoming these limitations, we systematically selected and validated a Human Action Video Database, consisting of 768 individual video clips from 256 human action categories, with three exemplars each. To increase the efficiency of behavioral data collection and to avoid scene cuts, we aimed for 1 s video clips. To ensure that the chosen time window clearly depicted the action, we identified the most informative frame in each video using a deep neural network (ResNet-50[19]). Finally, we controlled for frame rate, resolution and aspect ratio. For details on the generation of our Human Action Video Database, see Methods, 'Human Action Video Dataset'.

### Odd-one-out task data and modeling procedure

To train a computational model that is capable of extracting dimensions reflecting human action recognition and understanding, we had to collect human behavioral data where humans rely on these dimensions. To this end, following Hebart et al.[7], we implemented a triplet odd-one-out task using Amazon Mechanical Turk (MTurk) (https://www.mturk.com/). In this task, participants were shown a random subset of three action video clips from our Human Action Video Database (Fig. 3a). They were instructed to indicate which of the three videos did not fit the other two videos (for instructions, see Supplementary Fig. 1). Three individual videos drawn from 768 videos results in over 75 million different possible combinations of triplets. This combinatory space was aimed to be representative of the actions encountered in the natural world, where each video in the triplet serves as a context for the other two videos. Consequently, the task acts as a minimal categorization task, where different aspects of daily action perception are highlighted through each sampled video triplet.

Altogether, we gathered 1,252,704 odd-one-out ratings from 6036 participants. The unique triplets of those ratings represent roughly 1.6% of all possible combinations, given the 768 individual videos in our Human Action Video Database (see Methods, 'Data collection', for details).

From this broad behavioral dataset, we extracted dimensions using the sparse positive similarity embedding (SPoSE) modeling procedure[7,23]. First, human rating behavior expressed through the gathered behavioral data was learned by the model, implemented as a computational graph (see Fig. 3b). Weights for 100 latent dimensions were randomly initialized over all 768 video stimuli. In each iteration of the learning procedure, a triplet is selected. This triplet is represented by the vectors in the model, representing the three videos. Second, the proximity between these three vectors is calculated, which is then translated into a choice probability. The model's choice is then compared against the ground truth, the actual participant's choice given the triplet. In each step of this optimization procedure, the weights are updated using error backpropagation. This approach is repeated for all available triplet ratings and therefore is able to capture the human choice process[7]. Note that the algorithm holds a positivity as well as a sparsity constraint (implemented as $L_1$-normalization). Therefore, the initial 100 latent dimensions are shrunk down iteratively to a smaller subset of dimensions, as dimensions showing summed weight values smaller than 0.1 are removed.

Using 779,467 trials, which surpassed filtering to secure overall data quality (see Methods, 'Data analysis', for details), we extracted a final set of 28 dimensions. For the modeling procedure, the sparsity parameter $\lambda$ was set to 0.008 ($\lambda = 0.008$). This hyperparameter, controlling the tradeoff between model sparsity and complexity, was chosen both based on the literature[7,23] and through additional hyperparameter optimization (see Methods, 'Data analysis').

### Behavioral similarity of observed actions can be captured with 28 dimensions

The behavioral data, consisting of 779,467 data points after data filtering, combined with a computational modeling procedure, revealed 28 dimensions underlying human action recognition.

Note that the number of dimensions not only depended on the behavioral input data, but also on the hyperparameter lambda (see Methods, 'Data analysis', and Supplementary Fig. 4), chosen for the optimization procedure[7]. These dimensions were sorted in descending order with respect to the sum of their weights across videos (with dimension 1 showing the highest summed weight; see Fig. 2b, far right, and Supplementary Table 2, left column).

Next, we aimed to determine what is represented by each of these dimensions. To address this question, we explored the $28 \times 768$ dimension space in two different ways: (1) in a vertical fashion, identifying each individual dimension through its top-ranking video, and (2) in a horizontal fashion, identifying which dimensions are expressed within each video.

To gain an intuitive understanding of the 28 dimensions, we first visualized the videos with the highest weights for a given dimension (i.e., from top to bottom in the embedding model depicted in Fig. 3b, far right). Figure 4 shows a representative subset of dimensions, together with static images of the eight top-ranking videos. Visual inspection provides a first intuition regarding the kind of information captured by these dimensions, such as 'craft' (dimension 1), 'sport' (dimension 2) and 'food' (dimension 5). To be able to provide labels for each of these dimensions in an objective way, we asked a naïve group of participants ($N = 21$) to name each of the

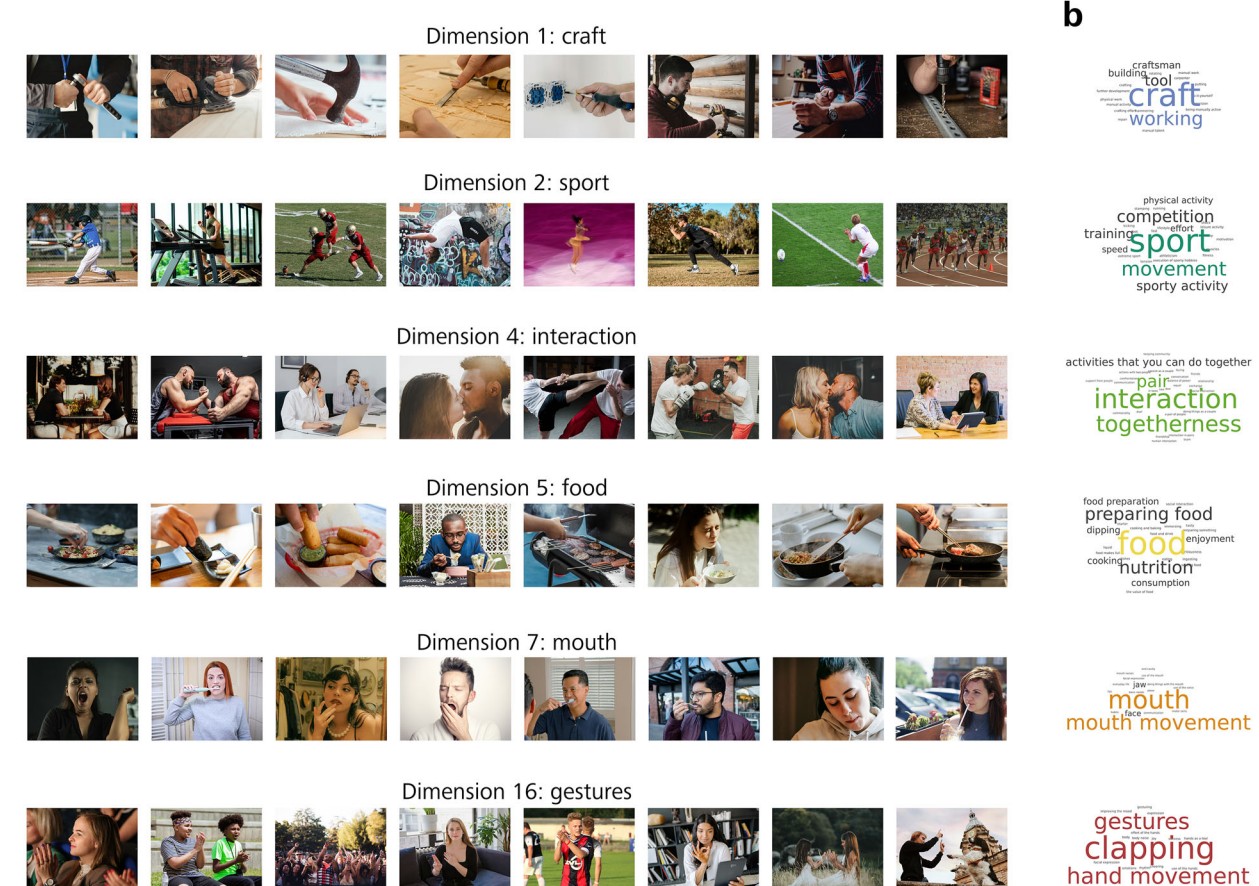

**Fig. 4 | Top rating stimuli and word clouds of example dimensions. a** Six representative dimensions were selected from all 28 action embedding dimensions. Each dimension is represented by static images of the top eight videos, showing the highest weights for the respective dimension. Dimension names were assigned by visually inspecting the top-rated videos. Note that the depicted static frames can only partially convey the dynamic information provided in the videos. **b** Labels provided by a group of $N = 21$ participants, after visually examining the dimensions through their corresponding videos (see also Fig. 2a, depicting the scale both used for the dimension naming and rating task), visualized as word clouds, where a larger font size corresponds to a higher naming frequency. For an overview of word clouds for all 28 dimensions and corresponding labels, see Supplementary Fig. 2 and

Supplementary Table 2. For this figure, the original MiT video frames were replaced by images under a free to use license (Pexels: Alexa Popovich, Andres Ayrton, Daniel Smyth, Gustavo Fring, Ksenia Chernaya, Luis Quintero, Mart Production, RDNE Stock project, Seher Dogan, Yan Krukau; Pixabay: 12019, 7721622, bbolender, brenkee, Fitnessstore112, Marvin_SNCR, PatrickBlaise, slavoljubovski, StockSnap, YasDO; Unsplash: Amy Hirschi, Bailey Alexander, Bernard Hermant, Debashis RC Biswas, Dominik Scythe, Donovan Sikaona, Gabriel Meinert, Giorgio Trovato, Hendo Wang, Jainam Sheth, James Kovin, Joe McFerrin, Julia Tsukurova, Keith Johnston, Kevin McCutcheon, Leah Hetteberg, Louis Hansel, Mark Adriane, Mary West, Michael Kahn, National Cancer Institute, Paul Trienekens, Rafaëlla Waasdorp, Rithwick. pr, Rod Long, Sander Sammy).

extracted dimensions. For this task, each dimension was visualized through a layout including a rating scale on which the dimension's representative videos were sorted according to their weights along the dimension (see also Fig. 2a, depicting the scale both used for the dimension naming and rating task, and Methods, 'Data collection'). In Fig. 4b, the resulting labels are visualized as word clouds, where naming frequency is indicated via font size (for word clouds of all 28 dimensions, see Supplementary Fig. 2). In addition, we extracted dimension labels from a large language model fed with the labels obtained from the dimension naming experiment (see Methods for details). A complete list of these dimensions, sorted in descending order by the sum of their weights among videos, together with the corresponding labels, is shown in Supplementary Table 2. As can be seen, the dimensions capture a range of different properties, ranging from properties related to goal-directed actions (e.g., craft, speaking, and sport) over properties related to living (e.g., crowd, child, and mouth) and non-living things (e.g., food and vehicle) to properties related to the environment (nature/outdoor), substance (water) and force (see also Supplementary Table 3, 1st and 2nd row). Thus, these findings indicate that our ability to interpret and recognize actions may rely on a unique combination of dimensions that capture a range of different properties from various domains at varying levels of abstraction. We will return to this point in the Discussion.

**Actions are associated with unique dimension profiles**

Next, we aimed to determine the dimensions that characterize individual actions. To address this question, we explored the dimensionality space at the level of individual action videos (i.e., from left to right in the embedding model depicted in Fig. 3b, far right).

After the computational modeling procedure, each video can be thought of as a row vector of length 28. Therefore, each of the 768 videos shows its own expression over the 28 extracted dimensions. To visualize this representation, we chose a circular bar plot (or rose plot). In this visualization, all 28 dimensions are represented as individual petals, ordered circularly in a counterclockwise fashion. The size of each petal reflects the extent to which each dimension is represented in the depicted action video. In Fig. 5, a representative selection of twelve out of the original 768 action videos is depicted together with their corresponding rose plots. Note that for ease of readability, only the highest expressed dimensions are labeled.

As can be seen, each individual video is associated with a unique set of dimensions. As an example, while the videos shown in the second row of Fig. 5 are characterized by a high weight on the 'sport' dimension, they differ with respect to the expression of other dimensions (e.g., relaxation, water, or vehicle). Hence, different dimension profiles are created for actions like 'balancing', 'boating' or 'bicycling', although they all contain at least one

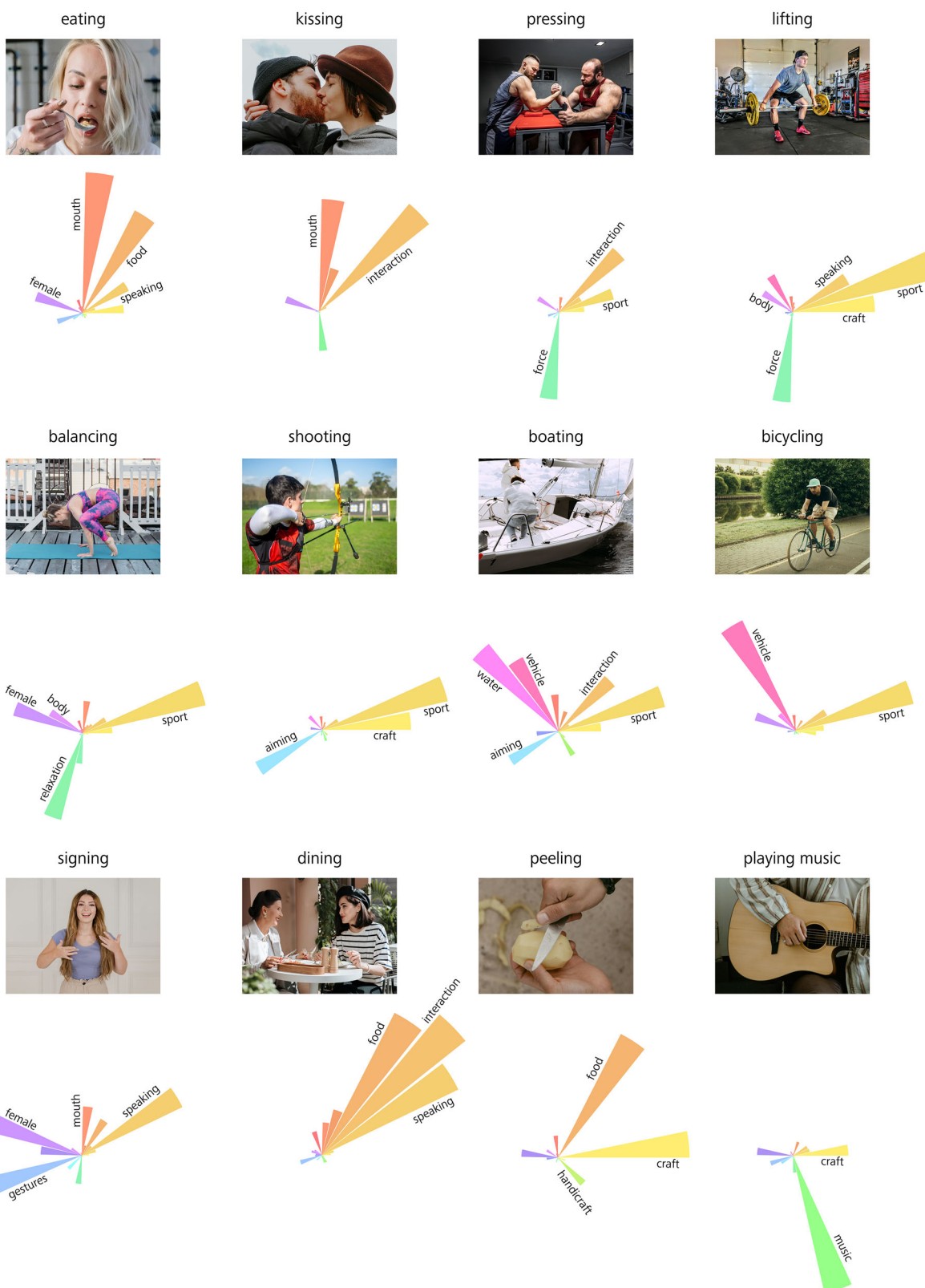

**Fig. 5 | Example actions and their corresponding dimensions are visualized through rose plots.** Depiction of twelve individual video stimuli together with their corresponding circular bar plots (or rose plots), expressing actions through their corresponding dimensions. Within each rose plot, the size of each petal represents the contribution of each individual dimension. The petals are ordered in a counterclockwise fashion, starting from three o'clock (dimension 1–28). For a better visualization, only the largest petals were labeled with their corresponding dimension names. For this figure, the original MiT video frames were replaced by images under a free-to-use license (Pexels: Alexa Popovich, cottonbro studio, Kampus Production, Maksim Goncharenok, Mart Production, Marta Wave, RDNE Stock project; Unsplash: Sam Sabourin, Toa Heftiba).

shared dimension. Likewise, the human action 'eating' can be characterized by the combination of the dimensions 'mouth' and 'food', while the combination of the dimensions 'mouth' and 'interaction' corresponds to the action 'kissing'. Together, these results suggest that the recognition of actions relies on a unique set of dimensions, where the similarity between different actions can be quantified based on the similarity of the dimension profile associated with each action. We will return to the interpretation of these results in the discussion.

## Human typicality ratings along dimensions correlate with dimension weights

Next, we aimed to determine whether the dimensions were just an effect of the SPoSE modeling procedure or whether participants actually use these dimensions to make their action similarity judgements. To address this challenge, we conducted a dimension rating experiment (see Methods, 'Data collection' for details). $N = 21$ naïve participants were asked to rate a randomly selected subset of 20 action videos among all 28 dimensions. Each dimension was represented through a continuous rating scale (see Fig. 2a). The task was to rate how typical or atypical the reference video was on the dimension scale, represented through its corresponding videos. From these ratings, we calculated a similarity matrix (see Fig. 2b, left). This matrix was then compared against a similarity matrix based only on the SPoSE model weights gained from the computational modeling procedure (see Fig. 2b, right). A comparison of these two matrices (using Pearson correlation) showed a high correlation of 0.78 ($P < 0.001$; 95% CI, 0.71–0.83, see Methods, 'Data analysis' for details). These results demonstrate that the dimensions revealed by the SPoSE modeling procedure indeed can be used by naïve participants without explicit instructions regarding the content of these dimensions. We will return to the implications of this observation in the discussion.

## Discussion

This study aimed to identify the principles underlying our ability to classify actions performed by others despite the variability in how these actions are executed. We argued that actions, like objects, can be represented in a multidimensional space, where each dimension reflects psychologically meaningful distinctions (e.g., refs. 3,9,14). According to this view, insights into the representations that support action understanding can be gained by revealing the key dimensions underlying the multidimensional space of a wide range of actions (768 video stimuli from 256 different action categories, with three exemplars each). Specifically, we wished to determine (1) what these key dimensions represent and (2) if they are interpretable and meaningful to a degree that they are relevant for behaviorally derived action similarity judgements in a separate group of participants.

Using a computational model of perceived similarity judgments, we discovered 28 key dimensions describing human action recognition (see Supplementary Table 2). The dimensions revealed by the computational model were visualized through their corresponding top-ranking stimuli and labeled by a separate group of participants. The dimensions captured different types of information from broadly different domains, including human actions, living and non-living things, environment, substance, and force, at varying levels of abstraction (see also Lingnau & Downing[3] for a recent discussion). Moreover, we obtained a close correspondence between dimension weights from the computational model and explicit ratings of actions along the dimensions by a naïve cohort of participants, underlining the dimension's interpretability. These results provide meaningful advances in understanding the kind of representations that support action understanding. In the following sections, we will discuss the content of these dimensions, their relation to previous studies, the theoretical importance of our results, future directions, and limitations of the current study.

Most dimensions (16 out of 24) were obtained within the broader domain 'human actions' (e.g., Craft, Sport, Speaking, and interaction). Seven out of 24 dimensions are associated with the broader category 'living things' (e.g., Mouth, Crowd, Child, and Female). The dimension 'crowd' might be related to actions typically performed in groups, while the dimensions 'child' and 'female' might reflect information about age- and gender-stereotypical actions and their agents. The remaining dimensions were spread across non-living things (Food, Vehicle; 2 out of 28), and environment, substance, and force (1 dimension each). Most dimensions with higher summed weights correspond to broad categories at a higher, more abstract level from the domain 'human actions' (e.g., Craft and Sport), living things (e.g., Crowd and Child), non-living things (Food, Vehicle), and environment (Nature/Outdoors). By contrast, most dimensions with lower weights correspond to more specific information at a lower level within the broad categories 'human actions' (e.g., cleaning, knotting, aiming, drawing, and winter sports), living things (e.g., body and foot), substance, and force.

Some dimensions were related to similar content, but at varying levels of abstraction (e.g., sport, winter sports; interaction, gestures; craft, handicraft). One advantage of dimensions related to similar content at varying levels of abstraction may be that they enable flexibility in adapting to the requirements of the current task and the context in which an action is processed. In line with this view, several recent behavioral and neuroimaging studies demonstrated that the taxonomic level at which an action is processed is reflected in the exposure duration that is required to recognize an action[15,25], the kind of features that are listed by human participants[15], and the underlying neural activation patterns[26].

Care needs to be taken when interpreting individual dimensions revealed by the SPoSE modeling procedure, given that the number of dimensions depends on the hyperparameter lambda (see Methods, 'Data analysis' for details), and that the dimensions may be correlated with each other, such that additional dimensions not explicitly revealed by our approach could be coded in all other dimensions. That said, the final dimensions are either extremely sparse or contain values in a reasonable range, providing the dimensionality of the solution for free and obviating the need to select a criterion for selecting the number of dimensions. Moreover, the number of dimensions was relatively stable across different random initializations of the weight matrix used to initialize the SPoSE model (see Supplementary Fig. 3). The observation that a comparably low-dimensional solution is sufficient for capturing similarity judgments is in line with other recent work[7,21,27]. In fact, a comparison of the summed weight of the first 10 dimensions obtained in the current study, focusing on action recognition, and those revealed by Hebart et al.[7], focusing on object recognition, suggests that the percentages for the dimensions obtained in the current study are slightly higher than those obtained by Hebart et al.[7], which might be due to the lower number of dimensions obtained in the current study (see Supplementary Table 4 for details).

As shown in Supplementary Table 3, most dimensions reported by previous studies, using a range of approaches[9,10,12,13,17], can be integrated into the action space model revealed in the current study, speaking for its broad explanatory profile (for exceptions, see refs. 12,17). Dimensions that seem most consistent across these various studies are related to instrumental/goal-directed actions, food and food-related actions, and expressive/leisure actions. That said, in comparison to the results obtained from previous studies, the dimensions revealed in the current study correspond to a broader range of domains, including not only information related to human actions and manipulable non-living things (reported by most previous studies), but also living things, non-manipulable non-living things, environment, substance, and force. This clearly demonstrates the unique added value of our data-driven approach for disentangling the nature of action representations in humans. Importantly, the action typicality ratings along the 28 dimensions gathered from a separate group of participants revealed a close correspondence to the dimension information revealed by the SPoSE modeling procedure, suggesting that the dimensions indeed are meaningful and interpretable to a degree that they are relevant for the ability to recognize and categorize actions. These results connect information obtained from a large sample size back to the level of individual participants and further underline that the behavior-derived dimensions obtained in the current study help individuals categorize and interpret actions performed by other people.

Visualizations of each action video via rose plots (see Fig. 5) highlight that actions are represented by unique profiles along several dimensions (see also refs. 10,28,29). We hypothesize that the unique profile of an observed action is compared to a range of reference profiles stored in long-term memory, and that a label is assigned to the observed action based on the reference profile that resulted in the highest similarity with the profile of the observed action. Note that while some of the dimensions obtained in the current study may seem categorical (e.g., mouth and food), the dimensions revealed by the modeling procedure reflected graded responses, enabling us to quantify the degree to which these dimensions are expressed in dynamic actions. As we will discuss in more detail in the following section, this property has important implications for future lines of research.

The dimensions revealed in the current study provide the basis for a promising direction for future research regarding the behavioral and neural correlates of these dimensions. First, it will be important to determine which of the dimensions are most relevant to recognize and distinguish between actions. Second, we expect important insights into the computational goals of different brain regions by examining the degree to which, where and when the multidimensional action space model and individual dimensions underlying this model can explain neuronal activity (see also refs. 9,13,30), and how this activity contributes to the ability to recognize actions. Third, it will be interesting to determine whether some of the dimensions reflect large-scale organizing principles along the ventral, dorsal and lateral visual pathway (but see ref. 31), as recently proposed for dimensions such as object size[32], sociality and transitivity[33], and input modality[34] (see also ref. 11). Fourth, examining the stability of dimensions within and between participants is appealing, as it will give insights into the roles of expertise, knowledge, and personal development. Fifth, it will be interesting to determine the degree to which the multidimensional action space model changes over time (both short- and long-term), e.g., as a function of the task, experience, or training. To address this latter question, we expect important insights from examining the dynamics—at different temporal scales—underlying the representation of individual dimensions. As an example, visualizing an action's dimension profile through the petals of a rose plot (see Fig. 5), would the petal sizes of dimensions vary for certain frames of an action video, highlighting different aspects of an action at individual time points? Finally, another important direction for future research will be to integrate the dimensions revealed in the current study into models explaining causal relationships, like scripts[35,36] or action and knowledge frames[3,37].

## Limitations

As with any study, there are several limitations that warrant consideration. To generate stimuli suitable for the triplet odd-one-out task, we had to select a range of different action categories. Other studies choose their stimulus categories based on other source material, like the American Time Use Survey[9,38–41]. Our starting point was the MiT dataset, which includes 339 categories[18]. Although the category selection was based on common verbs and word frequencies[42,43] and clustering approaches[44,45], Monfort et al.[18] also included Amazon Mechanical Turk workers in the procedure of video annotation[18]. The selected cohort of AMT workers might have influenced the final naming of the MiT dataset. While carefully reducing the original 339 action categories to 256 (see Methods, section 'Human Action Video Dataset', for details), we maintained original category names (with minor adjustments, see Supplementary Table 1), potentially including naming biases from Monfort et al.'s annotation procedure[18]. Note that Monfort et al.[46] recently addressed the issue of multiple action video labeling with their Multi-Moments in Time dataset (M-MiT), paving the ground for future studies examining this point more systematically[46]. In sum, while we are aware of these potential caveats, our careful selection of action categories resulted in a unique Human Action Video Dataset that is large enough for the purposes of data-driven approaches while at the same time being controlled with respect to the underlying meta structure, typicality ratings and the ability to name these actions. We hope that this video dataset, which we plan to make available to the scientific community, will inspire future behavioral and neuroimaging studies.

## Conclusion

Taken together, using a fully data-driven approach, combining computational modeling with crowdsourcing of behavioral similarity judgements for a wide range of actions, we identified key dimensions corresponding to a broad range of domains, including human actions, non-living and living things, environment, substance, and force, at varying levels of abstraction. Importantly, we demonstrated the relevance of these dimensions for behavior-derived action similarity comparisons in a separate group of participants. Our results thus provide a promising perspective on the principles underlying our ability to recognize actions, supporting the view that actions can be depicted in a multidimensional space, and that the similarity between different actions can be quantified by their Euclidean distance in this space. Finally, the dimensions revealed in our study pave the way for future lines of behavioral and neuroimaging studies, testing predictions generated from the quantification of the similarity between actions based on their multidimensional action profile.

## Data availability

Data needed to evaluate the main conclusions in the paper is available via an OSF repository (https://osf.io/b79eu/).

## Code availability

The MATLAB code reproducing results reported in this paper is available via an OSF repository (https://osf.io/b79eu/).

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

## Acknowledgements
We would like to thank Lisa Kaiser, Lucca Scheuermeyer, Johannes Schmidt, Pauline Schlegel, Simon Michael Frenzel, Lilia Döhler, Katharina Frey and Julian Wieland for their help with stimulus and data collection. Moreover, we are thankful to Oleg Vrabie for his support on the most informative frame pipeline, to Robert Bosek for his assistance during model setup and testing, and to Marius Zimmermann for his feedback on the paper.

## Author contributions
A.B. conceived, designed and performed the experiments, analyzed the data and wrote the paper. M.N.H. conceived and designed the experiments, contributed analysis tools and wrote the paper. A.L. conceived and designed the experiments and wrote the paper.

## Funding

## Competing interests
The authors declare no competing interests.
