## [Transparent Peer Review file · Communications Psychology]

Revealing Key Dimensions Underlying the Recognition of Dynamic Human Actions

Corresponding Author: Professor Angelika Lingnau

Version 0:

Decision Letter:

Dear Professor Lingnau,

Thank you for your patience during the peer-review process. Your manuscript titled "Revealing Key Dimensions Underlying the Recognition of Dynamic Human Actions" has now been seen by 3 reviewers, whose comments are appended below. You will see that they find your work of some potential interest. However, they have raised quite substantial concerns that must be addressed. In light of these comments, we cannot accept the manuscript for publication, but would be interested in considering a revised version that fully addresses these serious concerns.

We hope you will find the Reviewers' comments useful as you decide how to proceed. Should additional work allow you to address these criticisms, we would be happy to look at a substantially revised manuscript. If you choose to take up this option, please highlight all changes in the manuscript text file, and provide a detailed point-by-point reply to the reviewers.

Editorially, we consider it crucial that the experimental paradigm and analysis procedure are appropriate for the study of action recognition. To this end, please thoroughly address the reviewers' concerns regarding the appropriateness of using the odd-one-out paradigm. Please also include additional model comparisons and empirical data if applicable.

I am attaching a checklist that details critical reporting requirements for the revised manuscript. Please attend to each item and ensure your manuscript is fully compliant. We are requesting that your manuscript aligns with these requirements as this facilitates the evaluation of your manuscript, reducing delays in re-review and potential future acceptance. If your revised manuscript is not aligned with these requests on major issues, such as those concerning statistics, it may be returned to you for further revisions without re-review. Additional information can be found in our style and formatting guide Communications Psychology formatting guide.

If the revision process takes significantly longer than five months, we will be happy to reconsider your paper at a later date, provided it still presents a significant contribution to the literature at that stage.

Please use the following link to submit your

- revised manuscript,
- point-by-point response to the referees' comments,
- cover letter (as a separate document),
- the Editorial Policy Checklist (see below),
- the Reporting Summary (see below), and

- the completed Editorial Request Table (attached):

Link Redacted

Thank you for the opportunity to review your work.

Best regards,

Troby Lui

Troby Lui, PhD
Associate Editor
Communications Psychology

REVIEWER EXPERTISE:

Reviewer #1: action observation

Reviewer #2: action observation

Reviewer #3: action, cluster analysis

REVIEWER REPORTS:

Reviewer #1 (Remarks to the Author):

This is a very interesting and well executed study exploring the representational space underlying natural action perception. The authors compile an extensive (768) set of 1 second duration videos of human actions from the 'Moments in time' database – a database containing ~.25M short video clips. Using the same approach as a previous study investigating a multidimensional representation of natural objects (Hebart et al., 2020), the authors identify 28 'meaningful dimensions' on which their actions vary. In a separate group of participants, they test whether the actions can be rated along the dimensions identified. They find a close correspondence between the magnitude of the dimensions determined from similarity data, and the explicit ratings of the dimensions.

Major points

Theoretical implications. I agree with the authors analysis, that they set out in the introduction, that we have yet to clarify the behaviourally relevant dimensional space underlying action representation. This currently stems from the use of different modelling methods and different stimuli (static/dynamic, naturalistic/controlled, visual actions/verbs etc.) used in studies in this area. This study, using a different method, provides another model of action representation, I am, however, somewhat unsure how the authors think this study then actually reconciles some of the identified discrepancies in dimensions between this and the prior studies (e.g. Dima et al. 2022; Kabulska & Lingnau, 2022; Vinton et al. 2023; Thornton & Tamir 2022; Tucciarelli et al. 2019 etc.). There are a few sentences in the discussion section about this matter, but given the prominence of this problem as set out in the introduction, I think the paper would benefit greatly from significantly more analysis of the theoretical implications of this new model with respect to the other models previously proposed. How and why do these models differ? I am not sure it is enough to say that it is because there are more actions used to construct this model compared to others, as it is not clear to me what specifically these extra actions bring over the other sets.

Behavioural relevance. Through the paper there is repeated reference to the dimensions identified being 'behaviourally relevant'. I think this phrase invokes the idea that the information from these 28 action dimensions guides our actual human behaviour – e.g. our responses to and interaction with the acting agents. However, what I think is actually meant, and is tested in the study, is whether the dimensions identified from one set of set of human data converges with dimension 'rating' data from a different set of observers. We don't currently know that these 28 dimensions actually drive 'human behaviour'. The authors argue in the introduction that: "previous attempts were insufficient to explain behavior and, therefore, provided only an incomplete characterization of behaviorally relevant properties.". However, I am not sure this study makes the step from first showing the structure of our action representation to then demonstrating that this particular structure underpins our behaviour with agents any more so than some of the prior studies referred to. For example, in this study, actions are rated against the dimensions, whilst in Dima et al 2023 participants label the dimensions of actions. This is slightly different methodology, but not really the qualitative leap inferred. Therefore, I think this reference to behavioural relevance needs to be removed through the paper, and replaced with a more specific/explicit terms related to for example, 'action similarity comparisons', 'dimension ratings' etc.

Determining the number of dimensions. The selection of 28 dimensions (as opposed to 27 or 29) seems largely arbitrary and based upon a few choices (e.g. removing embeddings with a summed value $<.1$, λ). Naturally, action representation can be more closely modelled with higher dimensionality but at the expense of a parsimonious model. Given the substantive difference in the number of dimensions of this model and prior models of action representation with many fewer dimensions (e.g. 3 - Tucciarelli et al. 2019; 6 – Thornton & Tamir 2022; 4 – Vinton et al. 2023; 5 – Tarhan et al 2020 etc.), it would be very helpful to have an analysis of what the different models offer to our understanding of action space. Relatedly, I couldn't find a rank ordered list of the named 28 different dimensions in order of importance/sum or weights as referred to on Page 10 – could this be provided somewhere so that we can see what these dimensions represent? Given that there will be a 'law of

diminishing returns' by adding dimensions to a representational space, it would also be useful to see a metric of the importance of these different dimensions, and this might facilitate comparisons between this 28 dimensional model and other dimensional models.

Interpretation of the dimensions. The 28 dimensions are referred to in the paper as action properties, where each action might express several of these dimensions/properties to varying degrees. On the whole the labels for the different dimensions appear to be largely action categories (e.g. crafting actions or sporting actions), whilst some dimensions might be considered as properties of actions (e.g. force). So there seems to be some confusion here as to what the 28 dimensions represent. Further, on page 10, the authors refer to some of the dimensions as "Basic level actions (e.g. drawing, cleaning)", and refer elsewhere to dimensions as actions. Are the dimensions largely action categories (I think they are based upon the few examples give), so why are some dimensions actually referred to as specific actions? As I mention above, a list of the 28 dimensions would help, and allow the reader to make their own interpretations of the dimensions. However, it would be helpful if the authors could explain what they think the dimensions 'are' and also clarify this use of terminology. Maybe a taxonomy of the 28 dimensions (perhaps similar to that used by the same lab in Kabulska et al 2022) would help?

Related to this interpretation issue, I struggled to really understand 'female' as an action dimension. I don't understand how an action can vary in terms of its "femality", perhaps they can be executed by females (or not females), but this is not a property of the action, rather the agent. I would also have thought that approximately half of actions would convey this dimension, but I gathered from the manuscript this was probably not the case. Can the authors please explain how this particular dimension fits within their overall interpretation of what the dimensions represent? I think the dimension 'Child' might have similar issues of interpretation and would also need addressing.

I was not very convinced by the value of the MDS scaling applied to the similarity measures of the dimension labels. First, I think we need to understand how representative the MDS ordination is of the data. A stress plot of different dimensional solutions would help the reader evaluate whether the 28 dimensions really are best represented by these 2 dimensions and not 1, 3, 4, 5... Second, I found the labels given to the ends of the 2 dimensions in Figure 4 a little unconvincing when I looked at the distribution of the dimension labels. For example, why would 'water' be the most body-related dimension? Also, neuroimaging data (Wurm et al. 2017) indicates that a major structural division in the neural representation of different actions is between social actions and transitive actions. However, the cognitive organisation described by the MDS ordination in Figure 4 groups person-related action dimensions and transitive action dimensions together at the same end of one of the 2 MDS dimensions. Perhaps in a higher dimensional space transitivity and person-related labels might separate? In the discussion section on Page 22 "(1) transitive (person-related) vs..." seems to intimate that transitive actually refers to person-related actions, but that is not the meaning of 'transitive'.

I think to resolve these issues, that there needs to either further exploration of higher dimensional representations along with appropriate Stress/Shepard plots and better analysis of what these representations actually tells us about the organisation of these different action dimensions/categories. Currently, the statement in the discussion: "highlighting that the dimensions can be understood at multiple levels of explanation, thus integrating disparate perspectives onto action representations into a unifying framework." Is very vague and doesn't really provide much meaning or in-depth analysis of this lower-dimensional organisation. I think significantly deeper interpretation would be needed to fully justify the value of its inclusion. An alternative, would be to drop this aspect of the overall study entirely from the manuscript, as all the other interesting findings do not rely on this particular analysis.

Minor points

In the analysis between the 28-dimension action model and prior models in the discussion section, there are a couple of inaccuracies. To my knowledge (until this study) the maximum number of actual videos of actions used in similar studies was 240 (Vinton et al. 2023), however, Thornton & Tamir 2022 had (depending on the approach they took) for example 1875 verbs and 1729 nouns to generate their 'FAST-ACT' taxonomy. In addition, dimension 21 (force) is not a "new" dimension as it was one of the primary dimensions identified by Vinton et al. 2023 (formidableness).

Can you clarify in the legend for Figure 1 whether the large white "HCA" superimposed on the figure refers to Hierarchical Clustering Analysis.

Could you provide more information on the Stimulus Validation procedure where participants rate 'typicality'? What exactly was the information and instructions given to the participants? I don't think I could fully understand exactly what was done from the information given in this section.

Reviewer #2 (Remarks to the Author):

This study aimed to define the dimensions that comprise a multidimensional space describing actions using ratings of videos, AI, and computational modeling. The work is of potentially high significance given the need for a useful set of video action stimuli for behavioral and imaging research. The use of numerous stimuli and a large cohort of raters positively impacts the reliability and rigor of the approach. Although the sparse positive similarity embedding modeling procedure is likely to be unfamiliar to most, the investigators have done a good job of explaining it relatively clearly.

Despite these strengths, the inspection of the labels provided by a large-language model for the resultant 28 dimensions suggests that there may be a fundamental difficulty with the behavioral method employed (odd-one-out paradigm) because participants' groupings (and labels assigned by a large-language model) are focused on broadly different types of information present in the videos. Some of that information does not seem to be action- (verb) related, but may instead focus on the objects/people (nouns) present in the videos. For example, one dimension is labeled "food", another is "female", a third is "animals" and a fourth is "vehicle". Some of the noun labels appear to reflect broad categories at a more abstract event or scene level, such as "crowd", "nature", and "interaction". On the other hand, some dimensions seem more clearly (lower-level) action-related, such as "aiming", "entering", and "speaking". The mixture of verbs (actions), nouns at a relatively basic level, and nouns at a higher, more abstract level raises an important concern that the instructions provided to raters

were not sufficiently focused on action recognition, per se. Had the instructions specified that participants should focus on recognizing the actions in the scenes (rather than who was doing the actions or in what context) it is quite possible that the results would have differed considerably. It is recognized that such specific instructions might seem counter to the investigators' goal to ferret out implicitly important dimensions, but the alternative problem (that participants were not always making judgments based on actions) is more fundamental. Alternatively or in addition to explicit instructions, participants could be encouraged to refrain from basing their judgments on broad scene, event, or agent-focused information by deliberately selecting video triplets in which two of the three reflect the same actions performed with different objects, by different agents, in different contexts (for example, pouring milk, detergent, or gasoline, performed by man, woman or child, in kitchen, laundry, or service station). It is recommended that additional data is collected with these considerations in mind to quell any concerns that the instructions and task may have biased the results.

The labels for the primary axes depicted in Figure 4 do not intuitively fit the dimensions shown. How were the labels derived? The dimensions form a triangle shape that instead could plausibly be characterized as organized with respect to effectors (leg/body on left, arm/hand on top, mouth/face on right). Additionally, the methods for the derivation of just two axes rather than three from the hierarchical clustering data is not well-explained.

It seems that some of the actions represented were represented in two or three videos (e.g., "kissing") while others seem to have appeared in only one video. The impact of this uneven redundancy on participants' performance is unclear.

Some of the 28 dimensions retained by the model seem very similar to one another (sport, sport/locomotion, and winter sports; craft and handicraft) such that it is unclear why the model retained them as separate dimensions. This should be discussed.

Reviewer #3 (Remarks to the Author):

This study aimed at revealing dimensions underlying action recognition. They selected 768 one-second video clips from 256 action categories. In their multistep selection process they tried to span a wide range of human actions and be less biased to specific categories like sports. Then employing the data-driven approach (sparse positive similarity embedding; SPoSE), they found 28 dimensions. That were later examined for interpretability.

After studying the dimensions underlying object recognition, the logical step was moving toward action recognition and using videos which involved more challenges.

- video selection and data collection involved multiple steps described in method section with details. However, adding diagrams or flowcharts in method section could be helpful in summarizing the whole process.
- to extract the main dimensions of action space, SPoSE is employed. What are the alternative approaches? How are the results robust following those different approaches?
- Fig.4 depicts results of multi-dimensional scaling. Please indicate how much of the data variance is captured by these two dimensions?
- The study follows the methods employed previously to determine main dimensions underlying object recognition. Then, it would be nice to compare the results for object and action recognition regarding number of dimensions, the amount of variance explained by the first few dimensions, taxonomic level of dimensions,

EDITORIAL POLICIES

We ask that you ensure your manuscript complies with our editorial policies and reporting requirements.

To that end, we require revised manuscripts to be accompanied by two completed items: a reporting summary that collects information on study design and procedure, and an editorial policy checklist that verifies compliance with all required editorial policies

- <https://www.nature.com/documents/nr-reporting-summary.zip>>Nature Research Reporting Summary
- <https://www.nature.com/documents/nr-editorial-policy-checklist.pdf>>Editorial Policy Checklist

All points on the policy checklist must be addressed. Your revised manuscript can only be sent back to the referees if these checklists are completed and uploaded with the revision.

Notes: If you have submitted a Stage 1 Registered Report, Review, Primer, Comment, or Perspective you do not need to submit these forms. If you have already submitted these forms, you may disregard this request.

Version 1:

Decision Letter:

Dear Professor Lingnau,

Your manuscript titled "Revealing Key Dimensions Underlying the Recognition of Dynamic Human Actions" has now been seen by our reviewers, whose comments appear below. In light of their advice I am delighted to say that we are happy, in principle, to publish a suitably revised version in Communications Psychology.

We therefore invite you to revise your paper one last time to address the remaining concerns of our reviewers and a list of editorial requests. At the same time we ask that you edit your manuscript to comply with our format requirements and to maximise the accessibility and therefore the impact of your work.

EDITORIAL REQUESTS:

SUBMISSION INFORMATION:

OPEN ACCESS:

At acceptance, you will be provided with instructions for completing the open access licence agreement on behalf of all authors. This grants us the necessary permissions to publish your paper. Additionally, you will be asked to declare that all required third party permissions have been obtained, and to provide billing information in order to pay the article-processing

charge (APC).

* **DATA AVAILABILITY:**

Link Redacted

Best regards,

Troy Lui

Troy Lui, PhD
Associate Editor
Communications Psychology

REVIEWERS' COMMENTS:

Reviewer #1 (Remarks to the Author):

Thank you for your considered replies to the points I raised in my original review. And thank you for your efforts in making changes to the manuscript in response.

Some additional points:

1. In your reply you explain that you make the changes in the discussion to more accurately state: "in comparison to the results obtained from previous studies, the dimensions revealed in the current study correspond to a broader range of domains, including not only information related to human actions and manipulable non-living things (reported by most previous studies), but also living and non-living things, environment, substance, and force."

Given this clarification that the 28-dimension model pertains to human actions + other relevant domains, then I think this clarification now needs reflecting in the abstract somewhere.

2. In your changes to the manuscript you now state: "Note that while some of the dimensions obtained in the current study may seem categorical (e.g. mouth, food), the dimensions revealed by the modeling procedure reflected graded responses, enabling us to quantify the degree to which these dimensions are expressed in dynamic actions. As we will discuss in more detail in the following section, this property has important implications for future lines of research." Although the modelling procedure may produce graded responses, I am still struggling to see where the evidence is that all your "dimensions" are continuous (as is implied by the use of the terminology: dimension, and your argument here). I imagine if you plot a frequency histogram of actions (from your stimulus set) in bins of different magnitude values for each of the 28 dimensions, for many of these dimensions you will not see a normal distribution of these values. Nor perhaps something approximating a normal distribution, Gaussian distribution or indeed even a flat distribution. I suspect for some of these 'dimensions' you will have the vast majority of actions falling within the lowest magnitude response bin, and a few actions that fall within the higher magnitude bins. Perhaps the authors could check this analysis and reflect on whether all your 'dimensions' are actually continuous dimensions, (within none being better defined as categories) as you claim. As currently, without evidence, I am still struggling to be convinced that a 'dimension' such as food is something on which all human actions vary – we would here be claiming that all human actions are to a greater or lesser extent related to food. Intuitively I expect that some are, but most human actions have nothing at all to do with food. This argument also applies to some other stated dimensions (e.g. sport, mouth, hiking, music etc.). If it turns out that the 'dimensions' are not all continuously varying, and a

few show distributions commensurate with a more appropriate term: categories (as I suspect) – then a sentence or two in the discussion acknowledging this point would be beneficial.

3. Thank you for Extended Data Table 3 – I think this really helps to place the study's results in the context of other recent relevant literature.

4. Thank you for all your other comments and changes, I think they are really helpful in clarifying your study for the reader.

Reviewer #2 (Remarks to the Author):

The authors appear to have satisfactorily addressed all of the reviewers' comments.

Reviewer #3 (Remarks to the Author):

I have reviewed the revised manuscript and am satisfied that the revisions have improved the manuscript. I have no further concerns and recommend the paper for publication.

Reviewer #1 (Remarks to the Author):

This is a very interesting and well executed study exploring the representational space underlying natural action perception. The authors compile an extensive (768) set of 1 second duration videos of human actions from the 'Moments in time' database – a database containing ~.25M short video clips. Using the same approach as a previous study investigating a multidimensional representation of natural objects (Hebart et al., 2020), the authors identify 28 'meaningful dimensions' on which their actions vary. In a separate group of participants, they test whether the actions can be rated along the dimensions identified. They find a close correspondence between the magnitude of the dimensions determined from similarity data, and the explicit ratings of the dimensions.

Major points

[1] Theoretical implications. I agree with the authors analysis, that they set out in the introduction, that we have yet to clarify the behaviourally relevant dimensional space underlying action representation. This currently stems from the use of different modelling methods and different stimuli (static/dynamic, naturalistic/controlled, visual actions/verbs etc.) used in studies in this area. This study, using a different method, provides another model of action representation, I am, however, somewhat unsure how the authors think this study then actually reconciles some of the identified discrepancies in dimensions between this and the prior studies (e.g. Dima et al. 2022; Kabulska & Lingnau, 2022; Vinton et al. 2023; Thornton & Tamir 2022; Tucciarelli et al. 2019 etc.). There are a few sentences in the discussion section about this matter, but given the prominence of this problem as set out in the introduction, I think the paper would benefit greatly from significantly more analysis of the theoretical implications of this new model with respect to the other models previously proposed. How and why do these models differ? I am not sure it is enough to say that it is because there are more actions used to construct this model compared to others, as it is not clear to me what specifically these extra actions bring over the other sets.

<<< Our response: We would like to thank the reviewer for asking us to elaborate on the theoretical implications of the dimensions identified in the current study. This comment, combined with comments #3 and 4 below, made us realize that we made it too hard for the reader to appreciate the content of the 28 dimensions and the way they relate to dimensions revealed in previous studies. To overcome this limitation, we applied the following changes to the manuscript. First, while the original submission already included a rank-ordered list of the 28 named dimensions in the order of importance/ summed weights in Extended Data Table 2, we realized that it is easy to miss this table. We therefore now refer to this table more explicitly at several crucial points in the manuscript (Results section, pages 12 and 14; Discussion, page 21; Methods, page 34). Second, following the suggestion in comment #4, we added an additional table (Extended Data Table 3), where we sorted the 28 dimensions into several broad domains. Third, using this taxonomy, we highlighted overlaps with previous studies using different stimulus material and analytic approaches. Several observations can be gained from this additional table:

(...)

As shown in Extended Data Table 3, most dimensions reported by previous studies, using a range of approaches (Dima et al., 2022; Kabulska & Lingnau, 2022; Thornton & Tamir, 2022; Tucciarelli et al., 2019; Vinton et al., 2022), can be integrated into the action space model revealed in the current study, speaking for its broad explanatory profile (for exceptions, see Thornton & Tamir, 2022, and Vinton et al., 2022). Dimensions that seem most consistent across these various studies are related to instrumental / goal-directed actions, food and food-related

actions, and expressive / leisure actions. That said, in comparison to the results obtained from previous studies, the dimensions revealed in the current study correspond to a broader range of domains, including not only information related to human actions and manipulable non-living things (reported by most previous studies), but also living and non-living things, environment, substance, and force. This clearly demonstrates the unique added value of our data-driven approach for disentangling the nature of action representations in humans. Importantly, the action typicality ratings along the 28 dimensions gathered from a separate group of participants revealed a close correspondence to the dimension information revealed by the SPoSE modelling procedure, suggesting that the dimensions indeed are meaningful and interpretable to a degree that they are relevant for the ability to recognize and categorize actions. These results connect information obtained from a large sample size back to the level of individual participants and further underline that the behavior-derived dimensions obtained in the current study help individuals categorize and interpret actions performed by other people.

Visualizations of each action video via rose plots (see Fig. 4) highlight that actions are represented by unique profiles along several dimensions (see also Binder et al., 2016; Kabulska & Lingnau, 2022; Vinson & Vigliocco, 2008). We hypothesize that the unique profile of an observed action is compared to a range of reference profiles stored in long-term memory, and that a label is assigned to the observed action based on the reference profile that resulted in the highest similarity with the profile of the observed action. Note that while some of the dimensions obtained in the current study may seem categorical (e.g. mouth, food), the dimensions revealed by the modeling procedure reflected graded responses, enabling us to quantify the degree to which these dimensions are expressed in dynamic actions. As we will discuss in more detail in the following section, this property has important implications for future lines of research.

(...)

Conclusion. Taken together, using a fully data driven approach, combining computational modelling with crowdsourcing of behavioral similarity judgements for a wide range of actions, we identified key dimensions corresponding to a broad range of domains, including human actions, non-living and living things, environment, substance, and force, at varying levels of abstraction. Importantly, we demonstrated the relevance of these dimensions for behavior-derived action similarity comparisons in a separate group of participants. Our results thus provide a new perspective on the principles underlying our ability to recognize actions, supporting the view that actions can be depicted in a multidimensional space, and that the similarity between different actions can be quantified by their Euclidian distance in this space. Finally, the dimensions revealed in our study pave the way for future lines of behavioral and neuroimaging studies, testing predictions generated from the quantification of the similarity between actions based on their multidimensional action profile.

[2] Behavioural relevance. Through the paper there is repeated reference to the dimensions identified being 'behaviourally relevant'. I think this phrase invokes the idea that the information from these 28 action dimensions guides our actual human behaviour – e.g. our responses to and interaction with the acting agents. However, what I think is actually meant, and is tested in the study, is whether the dimensions identified from one set of set of human data converges with dimension 'rating' data from a different set of observers. We don't currently know that these 28 dimensions actually drive 'human behaviour'. The authors argue in the introduction that: "previous attempts were insufficient to explain behavior and, therefore, provided only an incomplete characterization of behaviorally relevant properties.". However, I am not sure this study makes the step from first showing the structure of our action representation to then demonstrating that this particular structure underpins our behaviour with agents any more so than some of the prior studies referred to. For example, in this study, actions are rated against the dimensions, whilst in Dima et al 2023 participants label the dimensions of actions. This is slightly different methodology, but not really the qualitative leap inferred. Therefore, I think this reference to behavioural relevance needs to be removed through the paper, and replaced with a more specific/explicit terms related to for example, 'action similarity comparisons', 'dimension ratings' etc.

<<< Our response: We agree that this wording may have invited misunderstandings. The idea is indeed that the dimensions identified in the current study are relevant to our ability to recognize actions, categorize actions, and identify features relevant for discriminating actions from each other. Following the suggestion of this reviewer, we removed the reference to behavioral relevance and replaced it with more specific/ explicit terms.

[3] Determining the number of dimensions. The selection of 28 dimensions (as opposed to 27 or 29) seems largely arbitrary and based upon a few choices (e.g. removing embeddings with a summed value $<.1$, lambda). Naturally, action representation can be more closely modelled with higher dimensionality but at the expense of a parsimonious model. Given the substantive difference in the number of dimensions of this model and prior models of action representation with many fewer dimensions (e.g. 3 - Tucciarelli et al. 2019; 6 – Thornton & Tamir 2022; 4 – Vinton et al. 2023; 5 – Tarhan et al 2020 etc.), it would be very helpful to have an analysis of what the different models offer to our understanding of action space. Relatedly, I couldn't find a rank ordered list of the named 28 different dimensions in order of importance/sum or weights as referred to on Page 10 – could this be provided somewhere so that we can see what these dimensions represent? Given that there will be a 'law of diminishing returns' by adding dimensions to a representational space, it would also be useful to see a metric of the importance of these different dimensions, and this might facilitate comparisons between this 28 dimensional model and other dimensional models.

<< We would like to thank the reviewer for raising these important points. Regarding the selection of 28 dimensions, the procedure uses sparsity based on an L1 norm, which drives some dimensions towards zero. The impact of the L1 norm is determined during cross-validation to identify the best performing model. While the cutoff for selecting dimensions may appear arbitrary, it is extremely clear cut: either dimensions are sparse and can disappear, or they are not. In other words, different to methods like principal component analysis, the selection of dimensionality comes for free without requiring a choice of cutoff._

In the revised manuscript, we now better clarify that the cutoff wasn't arbitrary. Specifically, we updated the corresponding section in the Discussion as follows:

Care needs to be taken when interpreting individual dimensions revealed by the SPoSE modeling procedure, given the number of dimensions depends on the hyperparameter lambda (see Methods, 'Data analysis' for details), and that the

dimensions may be correlated with each other, such that additional dimensions not explicitly revealed by our approach could be coded in all other dimensions. That said, the final dimensions are either extremely sparse or contain values in a reasonable range, providing the dimensionality of the solution for free and obviating the need to select a criterion for selecting the number of dimensions. Moreover, the number of dimensions was relatively stable across different random initializations of the weight matrix used to initialize the SPOSE model (see Extended Data Fig. 3). The observation that a comparably low-dimensional solution is sufficient for capturing similarity judgments is in line with other recent work (Hebart et al., 2020; Josephs et al., 2023; Kramer et al., 2023). In fact, a comparison of the summed weight of the first 10 dimensions obtained in the current study, focusing on action recognition, and those revealed by Hebart et al. (2020), focusing on object recognition, suggests that the percentages for the dimensions obtained in the current study are slightly higher than those obtained by Hebart et al. (2020), which might be due to the lower number of dimensions obtained in the current study (see Extended Data Table 4 for details).

Regarding the substantive difference in the number of dimensions of the model obtained in the current study and prior models, inspired by comment #1 of this reviewer, we added Extended Data Table 3, in which we sorted the 28 dimensions into broader categories (see also our response to comment 1 above). In this table, the ten most important dimensions (according to the summed weights) are highlighted in bold font. Several interesting insights can be gained from the first two rows of Extended Data Table 3:

- Most dimensions (16 out of 28) were obtained within the broader domain of human actions, but we also obtained a substantial number of dimensions within the broader category 'living things' (7 out of 28). The remaining dimensions were spread across non-living things (2 out of 28), substance, force, and environment (1 dimension each)
- The most important dimensions were obtained in the domain of human actions (craft, sport, speaking, interaction) and manipulable objects (food), followed by dimensions belonging to the broader category of living things (crowd, mouth, child), environment (nature/ outdoor) and non-manipulable objects (vehicle). What these dimensions have in common is that they tend to be more general.
- Dimensions with lower importance tended to correspond to more specific and concrete terms (e.g. aiming, knotting, handicraft, hiking), while covering the broader domains of living things, human actions, substance and force.

In the revised version of the manuscript, we elaborated the corresponding section in the Discussion as follows:

Most dimensions (16 out of 24) were obtained within the broader domain 'human actions' (e.g. Craft, Sport, Speaking, Interaction). 7 out of 24 dimensions are associated with the broader category 'living things' (e.g. Mouth, Crowd, Child, and Female). The dimension 'crowd' might be related to actions typically performed in groups, while the dimensions 'child' and 'female' might reflect information about age- and gender-stereotypical actions and their agents. The remaining dimensions were spread across non-living things (Food, Vehicle; 2 out of 28), and environment, substance, and force (1 dimension each). Most dimensions with higher summed weights correspond to broad categories at a higher, more abstract level from the domain 'human actions' (e.g. Craft, Sport), living things (e.g. Crowd, Child), non-living things (Food, Vehicle), and environment (Nature/ Outdoors). By

contrast, most dimensions with lower weights correspond to more specific information at a lower level within the broad categories ‘human actions’ (e.g. cleaning, knotting, aiming, drawing, winter sports), living things (e.g. body, foot), substance, and force.

Regarding the comment regarding a rank-ordered list of dimensions, we refer the reviewer to Extended Table 2, to which we now refer more explicitly at several crucial points in the manuscript (see also our response to comment #1 above).

Regarding the last point, to facilitate the comparison between the 28 dimensions revealed in the current study and other dimensional models, we listed similar or related dimensions revealed by previous studies (rows 3-8 in Extended Data Table 3). For further details on the interpretation of this table and the way we address these observations in the manuscript, see our detailed response to comment #1 above.

[4] Interpretation of the dimensions. The 28 dimensions are referred to in the paper as action properties, where each action might express several of these dimensions/properties to varying degrees. On the whole the labels for the different dimensions appear to be largely action categories (e.g. crafting actions or sporting actions), whilst some dimensions might be considered as properties of actions (e.g. force). So there seems to be some confusion here as to what the 28 dimensions represent. Further, on page 10, the authors refer to some of the dimensions as “Basic level actions (e.g. drawing, cleaning)”, and refer elsewhere to dimensions as actions. Are the dimensions largely action categories (I think they are based upon the few examples give), so why are some dimensions actually referred to as specific actions? As I mention above, a list of the 28 dimensions would help, and allow the reader to make their own interpretations of the dimensions. However, it would be helpful if the authors could explain what they think the dimensions ‘are’ and also clarify this use of terminology. Maybe a taxonomy of the 28 dimensions (perhaps similar to that used by the same lab in Kabulska et al 2022) would help?

<< We would like to thank the reviewer for the constructive feedback, which made us realize that referring to the dimensions as action properties and action dimensions might be misleading. What we wish to express is that these dimensions are relevant for our ability to judge the similarity between actions (from which we conclude that they are relevant for the ability to recognize and categorize actions). To avoid this confusion, we replaced ‘action properties’ with ‘properties’ and changed ‘action dimensions’ to ‘dimensions’/ ‘dimensions underlying the ability to judge the similarity of actions’ throughout the manuscript.

To clarify what the 28 dimensions represent, we followed the suggestion of this reviewer and sorted the dimensions into several broader domains (see Extended Data Table 3), which also facilitated the comparison with dimensions revealed in previous studies. For a more detailed response, see our reply to comments #1 and #3 above.

[5] Related to this interpretation issue, I struggled to really understand ‘female’ as an action dimension. I don’t understand how an action can vary in terms of its “femality”, perhaps they can be executed by females (or not females), but this is not a property of the action, rather the agent. I would also have thought that approximately half of actions would convey this dimension, but I gathered from the manuscript this was probably not the case. Can the authors please explain how this particular dimension fits within their overall interpretation of what the dimensions represent? I think the dimension ‘Child’ might have similar issues of interpretation and would also need addressing.

<< Our response: Thank you for asking us to elaborate on the interpretation of the dimensions ‘female’ and ‘child’. Interestingly, similar dimensions have been identified by Hebart et al. (<<feminine>>, <<baby>>) and by Dima et al. (<<child>>). To get a better understanding of the information carried by these dimensions, we provide visualizations of the top-ranking images (taken from the Moments in Time dataset; Monfort et al., 2020) for these two dimensions in Figure A1 below.

Figure A1: Top ranking stimuli and word clouds for the dimension ‘female’ (dimension # 13, top row) and ‘child’ (dimension # 9; bottom row). For both dimensions, (a) the five top ranking stimuli are depicted (from left to right) and (b) the corresponding word clouds are depicted.

Several observations can be obtained from Figure A1.

- (1) Female: While the gender of the actors depicted in the top-ranking images (Figure A1a, top row) are not known, they appear to be female based on visual cues. Moreover, several of the depicted actions appear to be stereotypically associated with female individuals (e.g. cheerleading, gymnastics). This is also evident from the responses provided by participants in the labelling task, summarized in the word clouds shown in Extended Data Figure 2: activity, balance, empowerment, energetic woman, expression, female cliché, feminine, femininity, greeting, gymnastics, hands, instructing, knocking, movement, outdated image of women, own activity, presenting, relaxation, rotation, self-determination, self-presentation, shaking, stereotypical images of women’s behavior, wagging, woman, women doing things.
- (2) Child: The top-ranking stimuli (see Fig. A1a) for the dimension ‘child’ all show toddlers, babies or small children, in line with the responses provided by participants in the labelling task (baby, child, child activity, childhood, childishness, education, growing up, infant, innocence, new beginning, scenes from a photo album of growing children, small, toddler, toddler snapshots, young, young age). They appear to engage in stereotypical actions like crawling, rocking, and sleeping. The repertoire of actions young children can perform is not only very limited, but also distinct from the way these actions are performed by adults. It therefore seems reasonable to assume that the dimension ‘child’ contributes to the recognition and categorization of observed actions.

Together, the highest-ranking images for the dimensions <<female>> and <<child>>, combined with the information gained from the labels provided by an independent group of participants, suggest that these dimensions carry information about actions and their agents that are associated with specific age groups and genders.

In the revised version of the manuscript, we now refer to these dimensions as follows (Discussion, page 21, last paragraph):

Most dimensions (16 out of 24) were obtained within the broader domain ‘human actions’ (e.g. Craft, Sport, Speaking, Interaction). 7 out of 24 dimensions are associated with the broader category ‘living things’ (e.g. Mouth, Crowd, Child, and Female). The dimension ‘crowd’ might be related to actions typically performed in groups, while the dimensions ‘child’ and ‘female’ might reflect information about age- and gender-stereotypical actions and their agents.

[6] I was not very convinced by the value of the MDS scaling applied to the similarity measures of the dimension labels. First, I think we need to understand how representative the MDS ordination is of the data. A stress plot of different dimensional solutions would help the reader evaluate whether the 28 dimensions really are best represented by these 2 dimensions and not 1, 3, 4, 5... Second, I found the labels given to the ends of the 2 dimensions in Figure 4 a little unconvincing when I looked at the distribution of the dimension labels. For example, why would ‘water’ be the most body-related dimension? Also, neuroimaging data (Wurm et al. 2017) indicates that a major structural division in the neural representation of different actions is between social actions and transitive actions. However, the cognitive organisation described by the MDS ordination in Figure 4 groups person-related action dimensions and transitive action dimensions together at the same end of one of the 2 MDS dimensions. Perhaps in a higher dimensional space transitivity and person-related labels might separate? In the discussion section on Page 22 “(1) transitive (person-related) vs...” seems to intimate that transitive actually refers to person-related actions, but that is not the meaning of ‘transitive’. I think to resolve these issues, that there needs to either further exploration of higher dimensional representations along with appropriate Stress/Shepard plots and better analysis of what these representations actually tells us about the organisation of these different action dimensions/categories. Currently, the statement in the discussion: “highlighting that the dimensions can be understood at multiple levels of explanation, thus integrating disparate perspectives onto action representations into a unifying framework.” Is very vague and doesn’t really provide much meaning or in-depth analysis of this lower-dimensional organisation. I think significantly deeper interpretation would be needed to fully justify the value of its inclusion. An alternative, would be to drop this aspect of the overall study entirely from the manuscript, as all the other interesting findings do not rely on this particular analysis.

<<< Our response: We would like to thank the reviewer for this constructive criticism of the MDS scaling approach. Given the limitations raised by the reviewer and the lack of deeper insights provided by this analysis, we followed the suggestion of this reviewer and dropped this aspect entirely from the manuscript.

Minor points

(1) In the analysis between the 28-dimension action model and prior models in the discussion section, there are a couple of inaccuracies. To my knowledge (until this study) the maximum number of actual videos of actions used in similar studies was 240 (Vinton et al. 2023), however, Thornton & Tamir 2022 had (depending on the approach they took) for example 1875 verbs and 1729 nouns to generate their ‘FAST-ACT’ taxonomy.

<<< Our response: Thank you for pointing out these inaccuracies. The corresponding sections in the discussion have been rewritten in response to several of the other comments raised by this and the other reviewers. We carefully checked that the revised version of the manuscript no longer contains these inaccuracies.

(2) In addition, dimension 21 (force) is not a “new” dimension as it was one of the primary dimensions identified by Vinton et al. 2023 (formidableness).

<<< Our response: Thank you for pointing out this inconsistency – we adjusted the manuscript accordingly and do no longer list ‘force’ as a new dimension.

(3) Can you clarify in the legend for Figure 1 whether the large white “HCA” superimposed on the figure refers to Hierarchical Clustering Analysis.

<<< Our response: The reviewer is correct – we adjusted the Figure caption to make this point more explicit.

(4) Could you provide more information on the Stimulus Validation procedure where participants rate ‘typicality’? What exactly was the information and instructions given to the participants? I don’t think I could fully understand exactly what was done from the information given in this section.

<<<Our response: To gather typicality ratings, participants took part in an online experiment (see Methods section, Stimulus validation, page 26 in the manuscript). During the task, they were presented with a static image depicting an action (corresponding to one of the one second videos) on the top of a browser page, together with the corresponding label (e.g. ‘aiming’; see Figure A2 below for an example). Participants were instructed to rate the depicted action on how typical it represented the named action (e.g. “On a scale from 1 (very untypical) to 7 (very typical): How typical is the action ‘aiming’ depicted here?”). The rating was executed while clicking one of seven buttons depicting a seven item Likert-scale, where one represents a very untypical and seven a very typical representation of the action stimulus for the named action. Participants could navigate back and forth within the SoSciSurvey Webpage using the buttons on the lower end of the page.

Figure A2: Example trial for the online typicality rating experiment.

In the revised version of the manuscript, we added the following explanation to make this point more explicit (page 28, 2nd paragraph):

To that aim, the stimulus was depicted on the top of the browser window. The depicted action was named in the instructions, and participants were asked to rate how well the action was depicted in the video on a scale from one ('very untypical') to seven ('very typical').

Reviewer #2 (Remarks to the Author):

This study aimed to define the dimensions that comprise a multidimensional space describing actions using ratings of videos, AI, and computational modeling. The work is of potentially high significance given the need for a useful set of video action stimuli for behavioral and imaging research. The use of numerous stimuli and a large cohort of raters positively impacts the reliability and rigor of the approach. Although the sparse positive similarity embedding modeling procedure is likely to be unfamiliar to most, the investigators have done a good job of explaining it relatively clearly.

<<Our response: We would like to thank the reviewer for this positive evaluation of our work.

[1] Despite these strengths, the inspection of the labels provided by a large-language model for the resultant 28 dimensions suggests that there may be a fundamental difficulty with the behavioral method employed (odd-one-out paradigm) because participants' groupings (and labels assigned by a large-language model) are focused on broadly different types of information present in the videos. Some of that information does not seem to be action- (verb) related, but may instead focus on the objects/people (nouns) present in the videos. For example, one dimension is labeled "food", another is "female", a third is "animals" and a fourth is "vehicle". Some of the noun labels appear to reflect broad categories at a more abstract event or scene level, such as "crowd", "nature", and "interaction". On the other hand, some dimensions seem more clearly (lower-level) action-related, such as "aiming", "entering", and "speaking". The mixture of verbs (actions), nouns at a relatively basic level, and nouns at a higher, more abstract level raises an important concern that the instructions provided to raters were not sufficiently focused on action recognition, per se. Had the instructions specified that participants should focus on recognizing the actions in the scenes (rather than who was doing the actions or in what context) it is quite possible that the results would have differed considerably. It is recognized that such specific instructions might seem counter to the investigators' goal to ferret out implicitly important dimensions, but the alternative problem (that participants were not always making judgments based on actions) is more fundamental. Alternatively or in addition to explicit instructions, participants could be encouraged to refrain from basing their judgments on broad scene, event, or agent-focused information by deliberately selecting video triplets in which two of the three reflect the same actions performed with different objects, by different agents, in different contexts (for example, pouring milk, detergent, or gasoline, performed by man, woman or child, in kitchen, laundry, or service station). It is recommended that additional data is collected with these considerations in mind to quell any concerns that the instructions and task may have biased the results.

>> Our response: We would like to thank the reviewer to ask us to elaborate on this important point. First, as already mentioned in response to comment #4 by reviewer #1, we realized that the label 'action dimensions' may have invited misunderstandings. What we wish to express is that these dimensions are relevant to judge the similarity between observed actions, from which we conclude that they are relevant for the recognition and categorization of actions. Different sources of information are likely to contribute to these judgements. As mentioned above, we

adjusted the wording throughout the manuscript, using the term ‘dimensions’ and ‘dimensions underlying the ability to judge the similarity of actions’ instead of ‘action dimensions’.

The reviewer is right in pointing out that the participants’ groupings are based on **broadly different types of information**, including human actions, animate and inanimate objects, environment, substance, and force, and that these dimensions vary with respect to the level of abstraction/ concreteness (see also Extended Data Table 3 in the revised version of the manuscript, and our response to comment #4 raised by reviewer #1).

Regarding the **task of the observer**, the comment made us realize that we may have invited a misunderstanding since we used an illustrative example from the object perception literature to introduce the triplet odd-one-out task (i.e., finding the odd one out among an apple, a banana and a car). We would like to stress that in the instructions we used in our experiment, we explicitly state that the three videos show three **actions**, and that the task is to select the odd-one-out. Specifically, as shown in Extended Data Fig. 1 (also shown in Figure A3 below), we used the following wording (bold font used in this response letter for the sake of clarity only):

*‘In each round, you will see three **videos of performed actions**. Two of them will be more similar to each other. The job is to select the odd-one-out by clicking on it.*

*Which **actions** are more similar to each other? Sometimes the decision is very difficult. There are 32 such rounds.*

Figure A3: Instructions used for the triplet odd-one-out online task (see also Extended Data Fig. 1).

Please note that the wording had to be concise (to be compatible with the requirements of an online crowdsourcing experiment) while **highlighting the importance of the actions**. Importantly, the instructions **neither mentioned the agents or the context**, but **exclusively the actions**. In sum, our instructions were **chosen in the way suggested by the reviewer** (i.e., focusing on the actions rather than the agent or the observer).

We agree that the task of the observer (e.g. focusing on the goals or the means of an action) may have an impact on the obtained dimensions, and that there might exist multiple distinct spaces, or hierarchically organized spaces (for a recent discussion, see e.g. Lingnau & Downing, 2024). While it will be interesting for future studies to compare multiple action spaces obtained from different tasks, we believe that this question goes beyond the purposes of the current study.

Finally, we would like to point out that previous approaches in the object categorization literature, which motivated the current study, revealed dimensions from various broad categories and different levels of abstraction, in line with the results obtained in the current study (see e.g. Hebart et al., 2020).

In the revised manuscript, we explicitly discuss this topic in the Discussion (page 21-22) as follows:

(...) The dimensions captured different types of information from broadly different domains, including human actions, living and non-living things, environment, substance, and force, at varying levels of abstraction (see also Lingnau & Downing, 2024, for a recent discussion). (...)

Most dimensions (16 out of 24) were obtained within the broader domain ‘human actions’ (e.g. Craft, Sport, Speaking, Interaction). 7 out of 24 dimensions are associated with the broader category ‘living things’ (e.g. Mouth, Crowd, Child, and Female). The dimension ‘crowd’ might be related to actions typically performed in groups, while the dimensions ‘child’ and ‘female’ might reflect information about age- and gender-stereotypical actions and their agents. The remaining dimensions were spread across non-living things (Food, Vehicle; 2 out of 28), and environment, substance, and force (1 dimension each). Most dimensions with higher summed weights correspond to broad categories at a higher, more abstract level from the domain ‘human actions’ (e.g. Craft, Sport), living things (e.g. Crowd, Child), non-living things (Food, Vehicle), and environment (Nature/ Outdoors). By contrast, most dimensions with lower weights correspond to more specific information at a lower level within the broad categories ‘human actions’ (e.g. cleaning, knotting, aiming, drawing, winter sports), living things (e.g. body, foot), substance, and force.

Some dimensions were related to similar content, but at varying levels of abstraction (e.g. sport, winter sports; interaction, gestures; craft, handicraft). One advantage of dimensions related to similar content at varying levels of abstraction may be that they enable flexibility in adapting to the requirements of the current task and the context in which an action is processed. In line with this view, several recent behavioral and neuroimaging studies demonstrated that the taxonomic level at which an action is processed is reflected in the exposure duration that is required to recognize an action (Reger et al., 2025; Zhuang & Lingnau, 2022) the kind of features that are listed by human participants (Zhuang & Lingnau, 2022), and the underlying neural activation patterns (Zhuang et al., 2023).

Moreover, we explicitly point out the need for future studies to examine the impact of the task on the multidimensional action space model in the discussion as follows (page 22):

Fifth, it will be interesting to determine the degree to which the multidimensional action space model changes over time (both short- and long-term), e.g. as a function of the task, experience or training.

[2] The labels for the primary axes depicted in Figure 4 do not intuitively fit the dimensions shown. How were the labels derived? The dimensions form a triangle shape that instead could plausibly be characterized as organized with respect to effectors (leg/body on left, arm/hand on top, mouth/face on right). Additionally, the methods for the derivation of just two axes rather than three from the hierarchical clustering data is not well-explained.

<< Our response: We would like to thank the reviewer for this constructive criticism. Following the suggestion of R1, which raised similar criticisms, we decided to remove the MDS scaling approach entirely from the manuscript.

[3] It seems that some of the actions represented were represented in two or three videos (e.g., “kissing”) while others seem to have appeared in only one video. The impact of this uneven redundancy on participants’ performance is unclear.

>> Our response: Unless we misunderstood the question, we assume that this comment is based on the example stimuli shown in Figures 3 and 5. We would like to clarify that there is no uneven redundancy in our dataset: the overall dataset contained 256 action categories, with exactly three exemplars per action category, as described in the Methods section.

To avoid misunderstandings, we made this point more explicit in the revised version of the manuscript as follows (Methods section, page 28, 1st paragraph):

After additional stimulus validation (see next section), a total of 256 action categories was selected as final category set. For each of those 256 action classes, three representative exemplars were hand-selected, resulting in a total of 768 stimuli. They were evenly distributed across action categories to reduce any possible representative redundancy.

[4] Some of the 28 dimensions retained by the model seem very similar to one another (sport, sport/locomotion, and winter sports; craft and handicraft) such that it is unclear why the model retained them as separate dimensions. This should be discussed.

>> Our response: We would like to thank the reviewer to ask us to discuss this point. As also pointed out in our response to comment #X3 by reviewer #1, our procedure uses sparsity based on an L1 norm, which drives some dimensions towards zero. The impact of the L1 norm is determined during cross-validation to identify the best performing model. While the cutoff for selecting dimensions may appear arbitrary, it is extremely clear cut: either dimensions are sparse and can disappear, or they are not. In other words, different to methods like principal component analysis, the selection of dimensionality comes for free without requiring a choice of cutoff.

In the revised manuscript, we now better clarify that the cutoff wasn’t arbitrary. Specifically, we added the following section to the Discussion (page 21, last paragraph):

Care needs to be taken when interpreting individual dimensions revealed by the SPoSE modeling procedure, given the number of dimensions depends on the hyperparameter lambda (see Methods, ‘Data analysis’ for details), and that the dimensions may be correlated with each other, such that additional dimensions not explicitly revealed by our approach could be coded in all other dimensions. That said, the final dimensions are either extremely sparse or contain values in a reasonable range, providing the dimensionality of the solution for free and obviating the need to select a criterion for selecting the number of dimensions. Moreover, the number of dimensions was relatively stable across different random initializations of the weight matrix used to initialize the SPoSE model (see Extended Data Fig. 3). The observation that a comparably low-dimensional solution is sufficient for capturing similarity judgments is in line with other recent work (Hebart et al., 2020; Josephs et al., 2023; Kramer et al., 2023). In fact, a comparison of the summed weight of the first 10 dimensions obtained in the current study, focusing on action recognition, and those revealed by Hebart et al.

(2020), focusing on object recognition, suggests that the percentages for the dimensions obtained in the current study are slightly higher than those obtained by Hebart et al. (2020), which might be due to the lower number of dimensions obtained in the current study (see Extended Data Table 4 for details).

Moreover, as also pointed out in our response to comment #1 above, in the revised version of the manuscript, we now discuss that the model might retain dimensions that are similar to one another to enable flexibility. That is, depending on our task at hand, we need to process different types of information at different taxonomic levels (see e.g. Zhuang & Lingnau, 2022; Lingnau & Downing, 2024). As an example, we can tell based on the presence of a basketball and a basket that an actor is playing basketball, while information about body posture, kinematics and the opposed player is required if we need to distinguish between foul and fair play. A model capturing the ability to categorize actions should be able to support this ability.

Discussion, page 22, 3rd paragraph:

Some dimensions were related to similar content, but at varying levels of abstraction (e.g. sport, winter sports; interaction, gestures; craft, handicraft). One advantage of dimensions related to similar content at varying levels of abstraction may be that they enable flexibility in adapting to the requirements of the current task and the context in which an action is processed. In line with this view, several recent behavioral and neuroimaging studies demonstrated that the taxonomic level at which an action is processed is reflected in the exposure duration that is required to recognize an action (Reger et al., 2025; Zhuang & Lingnau, 2022) the kind of features that are listed by human participants (Zhuang & Lingnau, 2022), and the underlying neural activation patterns (Zhuang et al., 2023).

Reviewer #3 (Remarks to the Author):

This study aimed at revealing dimensions underlying action recognition. They selected 768 one-second video clips from 256 action categories. In their multistep selection process they tried to span a wide range of human actions and be less biased to specific categories like sports. Then employing the data-driven approach (sparse positive similarity embedding; SPoSE), they found 28 dimensions. That were later examined for interpretability.

After studying the dimensions underlying object recognition, the logical step was moving toward action recognition and using videos which involved more challenges.

[1] • video selection and data collection involved multiple steps described in method section with details. However, adding diagrams or flowcharts in method section could be helpful in summarizing the whole process.

<< Our response: We would like to thank the reviewer for this suggestion. Given that the manuscript currently already consists of five figures in the main section (including a flowchart illustrating the odd-one-out task and the modelling procedure; Fig. 2) and four figures in the Extended Data section, we hesitated to add an additional figure visualizing the video selection and data collection in the original submission. That said, we prepared a draft of such a figure (see Figure A4 below). Should the reviewer and/ or the editor have a strong preference for including such a figure in the manuscript, we are happy to follow this suggestion.

a**b****Figure A4 | ACTIONS-database creation steps.**

(a) Overview of the steps to create the ACTIONS-database from the Moments in Time (MiT) database (Monfort et al., 2020): (1) First, the MiT-database (highlighted green) served as a starting point for stimulus selection. From the over one million stimuli spanning 339 categories, a subset of approximately two thousand videos was selected. (2) Next, to identify the frame with the highest prediction accuracy for the correct category within each video (Most Informative Frame, MIF), these videos were fed into a deep neural network pipeline based on the ResNet-50 (He et al., 2015), finetuned on the MiT data. (3) The MIF was saved as a static image. Moreover, one second segments centered around the MIF were extracted from the original three second videos. (4) Additional quality control steps and behavioral stimulus validation experiments guided the final selection of stimuli for the ACTIONS dataset. Altogether 768 stimuli from 256 different action categories were gathered both as static images and as dynamic videos (highlighted green).

(b) Goals we aimed to reach during the creation of the ACTIONS database. (1) Aiming for a high representative power of the final dataset, not all the original 339 MiT categories were included. A hierarchical cluster analysis (HCA) using the softmax vectors from the ResNet-50 pipeline helped to identify (and to exclude) over-represented categories (a total of 83 categories were excluded). (2)

Starting from the MiT database, stimuli were not equally distributed over categories. Human selectors carefully selected exactly three representative stimuli per category. (3) Depending on the position of the MIF within the original MiT three second video, up to three different one second videos could be extracted. Human selectors carefully selected the most suitable of those up to three one second versions. This selection process additionally dealt with potential scene changes in the original MiT videos to maintain the highest quality and usability of the stimuli in the final ACTIONS dataset. (4) The original MiT videos differed in aspect ratio and resolution. To maintain homogeneity over stimuli in the ACTIONS database, these technical parameters were adjusted creating 480x360 pixels stimuli (landscape format), without padding or black bars.

[2] • to extract the main dimensions of action space, SPoSE is employed. What are the alternative approaches? How are the results robust following those different approaches?

<< Our response: Many alternative approaches could be employed, depending on the exact criteria one would like to gain from the dimensions. Currently, SPoSE and the variational Bayesian version of SPoSE, VICE, are, to our knowledge, the only methods that provide non-negative, sparse and continuous dimensions. Since the approach depends on an incomplete sample of similarities, other methods of low-rank matrix completion could be employed, based on which one could also run other methods, such as principal component analysis. However, given the goals of this study, we opted for sparse and non-negative dimensions, as derived from SPoSE. In the revised manuscript, we now better highlight the reason for choosing SPoSE (page 32, 2nd paragraph).

[3] • Fig.4 depicts results of multi-dimensional scaling. Please indicate how much of the data variance is captured by these two dimensions?

<< Our response: We want to thank the reviewer for his helpful remark. Given the limitations raised by the other two reviewers and the lack of deeper insights provided by this analysis, we removed the MDS analysis entirely from the manuscript.

[4] • The study follows the methods employed previously to determine main dimensions underlying object recognition. Then, it would be nice to compare the results for object and action recognition regarding number of dimensions, the amount of variance explained by the first few dimensions, taxonomic level of dimensions,

>> Our response: We would like to thank the reviewer for asking us to elaborate on this point. As also remarked by reviewers #1 and #2, the original version of the manuscript was lacking a more detailed elaboration of the information content of the 28 dimensions. Therefore, we decided to include an additional table (Extended Data Table 3) to clarify (a) the information content of the dimensions and (b) to compare the dimensions revealed in the current study with those reported in previous studies.

For a more detailed description of this table, the interpretation and the way we addressed this topic in the revised version of the manuscript, we refer to our responses to comments #1 and #3 by reviewer #1.

Regarding the comparison for object and action recognition regarding the number of dimensions and the amount of variance explained by the first few dimensions, we prepared an additional table (see Table A1 below, now shown as Extended Data Table 4 in the manuscript), where we provide the summed weights of the first 10 dimensions in relation to the total summed weights over all dimensions, producing percentage values of model weights for each dimension in the model underlying action recognition (obtained in the current study) and the dimensions underlying object recognition obtained by Hebart et al (2020). Overall, the percentages for the

dimensions obtained in the current study are slightly higher than those obtained by Hebart et al (2020), which might be due to the fact that the latter study revealed 21 dimensions more than the current study (49 compared to 28).

Table A1: Comparison of dimensions underlying action recognition (current study) and object recognition (Hebart et al., 2020) for dimensions 1-10. Columns 2 and 4 depict dimension labels, columns 3 and 5 show model weight percentages. Similar or very closely related dimensions that were obtained in both models within the top 10 dimensions are highlighted in bold font.

Dimension number	Action recognition (current study) 28 dimensions		Object recognition (Hebart et al., 2020) 49 dimensions	
	label	[%]	label	[%]
1	Craft	11.1	Made of metal / artificial / hard	8
2	Sport	10.7	Food-related / eating-related / kitchen-related	6.3
3	Speaking	7.6	Animal-related / organic	4.7
4	Interaction	5.2	Clothing-related / fabric / covering	3.4
5	Food	4.9	Furniture-related / household-related / artifact	3.3
6	Crowd	4.8	Plant-related / green	3.2
7	Mouth	4.7	Outdoors-related	2.8
8	Nature / Outdoor	4.5	Transportation / motorized / dynamic	2.8
9	Child	4.2	Wood-related / brown	2.8
10	Vehicle	4.2	Body part-related	2.7

We updated the discussion as follows (page 22, last paragraph):

(...) The observation that a comparably low-dimensional solution is sufficient for capturing similarity judgments is in line with other recent work (Hebart et al., 2020; Josephs et al., 2023; Kramer et al., 2023). In fact, a comparison of the summed weight of the first 10 dimensions obtained in the current study, focusing on action recognition, and those revealed by Hebart et al. (2020), focusing on object recognition, suggests that the percentages for the dimensions obtained in the current study are slightly higher than those obtained by Hebart et al. (2020), which might be due to the lower number of dimensions obtained in the current study (see Extended Data Table 4 for details).

Reviewer #1 (Remarks to the Author):

Thank you for your considered replies to the points I raised in my original review. And thank you for your efforts in making changes to the manuscript in response.

<< Our response: We would like to thank the reviewer for the careful, constructive review of our manuscript.

Some additional points:

1. In your reply you explain that you make the changes in the discussion to more accurately state: “in comparison to the results obtained from previous studies, the dimensions revealed in the current study correspond to a broader range of domains, including not only information related to human actions and manipulable non-living things (reported by most previous studies), but also living and non-living things, environment, substance, and force.” Given this clarification that the 28-dimension model pertains to human actions + other relevant domains, then I think this clarification now needs reflecting in the abstract somewhere.

<< Our response: Thank you for raising this point, we adjusted the abstract as follows:

This approach revealed 28 meaningful dimensions (e.g. interaction, sport and craft) which capture information concerning human actions as well as a broad range of related domains (e.g. living and non-living things).

2. In your changes to the manuscript you now state: “Note that while some of the dimensions obtained in the current study may seem categorical (e.g. mouth, food), the dimensions revealed by the modeling procedure reflected graded responses, enabling us to quantify the degree to which these dimensions are expressed in dynamic actions. As we will discuss in more detail in the following section, this property has important implications for future lines of research.” Although the modelling procedure may produce graded responses, I am still struggling to see where the evidence is that all your “dimensions” are continuous (as is implied by the use of the terminology: dimension, and your argument here). I imagine if you plot a frequency histogram of actions (from your stimulus set) in bins of different magnitude values for each of the 28 dimensions, for many of these dimensions you will not see a normal distribution of these values. Nor perhaps something approximating a normal distribution, Gaussian distribution or indeed even a flat distribution. I suspect for some of these ‘dimensions’ you will have the vast majority of actions falling within the lowest magnitude response bin, and a few actions that fall within the higher magnitude bins. Perhaps the authors could check this analysis and reflect on whether all your ‘dimensions’ are actually continuous dimensions, (within none being better defined as categories) as you claim. As currently, without evidence, I am still struggling to be convinced that a ‘dimension’ such as food is something on which all human actions vary – we would here be claiming that all human actions are to a greater or lesser extent related to food. Intuitively I expect that some are, but most human actions have nothing at all to do with food. This argument also applies to some other stated dimensions (e.g. sport, mouth, hiking, music etc.). If it turns out that the ‘dimensions’ are not all continuously varying, and a few show distributions commensurate with a more appropriate term: categories (as I suspect) – then a sentence or two in the discussion acknowledging this point would be beneficial.

<< Our response: We are grateful for the reviewer highlighting room for further clarification. We would like to point out that a specific distribution, such as Gaussian or flat, is not required for a graded response. Indeed, our dimensions were optimized for being sparse. This desirable property led to typical histograms with a long-tail decay function peaked at 0 and – depending on the dimension – a second mode with a distribution that resembles a Gaussian. For the example of food, this would mean exactly what the reviewer believes: Some objects are more

closely related to food, while others are less related, and most are completely unrelated. Please note the difference between a graded, continuous code, as claimed in the manuscript, and a dense code, as proposed by the reviewer. The attractive property of the identified dimensions is that they satisfy criteria from both feature-based models that propose binary properties as “dimensions” (e.g., “has arms”, “is thrown”, “is food”), and dimension-based models that propose continuous dimensions (e.g., “animate – inanimate”). By virtue of lying between these two extremes, sparse positive dimensions can be *both* at the same time, by reflecting the degree to which a property is expressed.

3. Thank you for Extended Data Table 3 – I think this really helps to place the study’s results in the context of other recent relevant literature.

<< Our response: Thank you for the feedback, and for the valuable suggestion that led to this table.

4. Thank you for all your other comments and changes, I think they are really helpful in clarifying your study for the reader.

<< Our response: Thank you – we appreciate the positive feedback!

Reviewer #2 (Remarks to the Author):

The authors appear to have satisfactorily addressed all of the reviewers' comments.

Reviewer #3 (Remarks to the Author):

I have reviewed the revised manuscript and am satisfied that the revisions have improved the manuscript. I have no further concerns and recommend the paper for publication.